# Tracing genetic diversity captures the molecular basis of misfolding disease

Pei Zhao[1,6], Chao Wang [1,2,6] ✉, Shuhong Sun[1,3,4], Xi Wang[1,5] & William E. Balch [1] ✉

Genetic variation in human populations can result in the misfolding and aggregation of proteins, giving rise to systemic and neurodegenerative diseases that require management by proteostasis. Here, we define the role of GRP94, the endoplasmic reticulum Hsp90 chaperone paralog, in managing alpha-1-antitrypsin deficiency on a residue-by-residue basis using Gaussian process regression-based machine learning to profile the spatial covariance relationships that dictate protein folding arising from sequence variants in the population. Covariance analysis suggests a role for the ATPase activity of GRP94 in controlling the N- to C-terminal cooperative folding of alpha-1-antitrypsin responsible for the correction of liver aggregation and lung-disease phenotypes of alpha-1-antitrypsin deficiency. Gaussian process-based spatial covariance profiling provides a standard model built on covariant principles to evaluate the role of proteostasis components in guiding information flow from genome to proteome in response to genetic variation, potentially allowing us to intervene in the onset and progression of complex multi-system human diseases.

Genome variation in the human population provides a vast reservoir for the protein fold to explore and evolve conformational plasticity required for form and function in changing complex biological environments- but also poses the risk for protein misfolding and/or aggregation that could lead to human genetic disease[1,2]. We now appreciate that many aspects of the evolving protein sequence-to-function-to-structure relationships are dynamically managed by a ubiquitous protein folding system referred to as proteostasis[3–6]. Proteostasis is composed of a collection of chaperone/co-chaperone components, trafficking components, degradation machineries, and signaling pathways such as the heat shock response (HSR) and unfolded protein response (UPR) that manage the fold in response to sequence variation and environmental stress across a lifespan[7–9].

The potential role of proteostasis in disease management is just beginning to be unraveled[10–15].

To address the central problem of information flow from the genome to the proteome in human biology in the context of the complex folding dynamics impacted by heat shock protein 90 (Hsp90) family members[16] that constitute a central ATP-dependent chaperone system in the cell, we developed a Gaussian process (GP) regression-based machine learning[1,17–21] approach to learn the basal state of the protein fold and its response to proteostasis management. GP can use a sparse collection of variants across the worldwide population and their associated phenotypes as a collective input to generate spatial covariance (SCV) maps of the response of every residue in the protein sequence and its residue–residue responses to environmental changes, including pharmacological intervention[1,17–21]. GP-SCV principled

[1]Department of Molecular Medicine, Scripps Research, La Jolla, CA, USA. [2]Present address: Institute of Systems and Physical Biology, Shenzhen Bay Laboratory, Shenzhen, China. [3]Present address: Department of Nutrition and Food Hygiene, Center for Global Health, School of Public Health, Nanjing Medical University, Nanjing, China. [4]Present address: Institute for Brain Tumors, Collaborative Innovation Center for Cancer Personalized Medicine, and Center for Global Health, Nanjing Medical University, Nanjing, China. [5]Present address: National Laboratory of Biomacromolecules, Institute of Biophysics, Chinese Academy of Sciences, Beijing, China. [6]These authors contributed equally: Pei Zhao, Chao Wang. ✉e-mail: chaowang@szbl.ac.cn; webalch@scripps.edu

modeling captures as a collective the variant changes that report on the evolutionary trajectory of the entire wild-type (WT) protein fold at atomic resolution that impacts health and disease of the host in response to the environment and natural selection[1,17–21].

To mechanistically dissect Hsp90 function on a single residue basis, we used a selective Hsp90 inhibitor PU-WS13 that specifically reduces the ATPase activity of the endoplasmic reticulum (ER) Hsp90 paralog GRP94, serving as a chemical mimetic of the role of co-chaperones that normally manage its ATPase cycle[22–29]. To trace the SCV principled dynamic profile of GRP94 ATPase activity on human variation, we examined its impact on the management of alpha-1-antitrypsin deficiency (AATD). AATD is caused by genetic variation in *SERPINA1* gene that encodes AAT, a prominent member of the serine protease inhibitor family[30]. AATD is an autosomal recessive inherited disease and the most prominent risk factor for the aging-related onset of chronic obstructive pulmonary disease (COPD) in response to smoking and airborne pollutants[30]. AAT is synthesized and secreted from hepatocytes for delivery to the lung at grams per day where it binds human neutrophil elastase (NE) and prevents NE-induced degradation of the extracellular matrix (ECM) in the lung[30,31]. Numerous variants in the AAT protein can cause protein misfolding during nascent synthesis in the ER, the first step of the secretory pathway[32–34], leading to aggregation with extended polymers that trigger liver disease phenotypes, such as fibrosis, cirrhosis and hepatocellular carcinoma[35–37]. AAT aggregation in the liver results in reduced secretion of functional AAT to the plasma, leading to a loss-of-function in the lung manifested as the inflammatory diseases emphysema, bronchitis and COPD[30,38]. AAT augmentation therapy using human plasma-derived purified AAT to re-supply the reduced level of AAT found in circulation has no impact on progression of liver aggregation disease and only modest impact on progression of lung disease[39]. The distinct liver and lung pathologies pose unprecedented challenges to develop therapeutics that preserve lung function while preventing polymer formation[32,40,41].

Herein, we utilize GP-SCV principled relationships to understand the impact of GRP94 ATPase activity on human variation affecting the level of intracellular or secreted AAT monomer and polymer pools, as well as the secreted NE inhibitory activity, based on 76 AAT variants found in the population that differentially contribute to disease onset and progression[32]. Strikingly, we find that the inhibition of GRP94 ATPase activity improves AAT monomer secretion and its NE inhibitory activity, while reducing both intracellular polymer load and polymer secretion for most pathogenic AAT variants, including the prominent Z-variant affecting 95% of the AATD population. GP-SCV principled relationships reveal the impact of GPR94 ATPase activity to (re)direct the N- to C-terminal cooperative folding of AAT to correct AATD clinical phenotypes impacting both liver and lung disease. We suggest that GP-SCV principled relationships provide a standard computational framework to address information flow from the genome to proteome affecting protein fold conversion by a broad range of proteostasis components affecting health and disease on a residue-by-residue basis[1,20,21,34], providing a precision approach to mitigate disease.

## Results
### Assaying the diverse folding and functional features of AAT variants
We recently showed that pharmacological activation of ATF6, a specific branch of the unfolded protein response (UPR) pathway that regulates the proteostasis environment in the ER, can improve the function of AAT variants while simultaneously reducing aggregation in the ER[34]. Preliminary results suggested that ER chaperones GRP78 and GRP94 are required for the restoration of function impacted by ATF6 activators[34]. As a paralog of the cytosolic Hsp90, GRP94 is a highly abundant ATP-dependent chaperone found in the ER[14,16,42–44]. It has

previously been proposed to direct the misfolded AAT protein to ER associated degradation (ERAD) pathway[45–47]. Like cytosolic Hsp90, how GRP94 recognizes and manages folding in response to genetic variation in the population, and how it could be modulated to manage ER misfolding diseases in response to variation impacted by ~33% of the protein load encoded by the human genome, remains unknown.

To probe the mechanism of how GRP94 manages the folding and function of AAT, we used AAT variants in the human population that harbor diverse folding conformations in the cell[32,33] including the most common Z-variant allele (E366K), leading to severe disease in 95% of the AATD population[41,48]. The collection of AAT variants contains 44 pathologic variants associated with AATD liver and lung phenotypes reported in the literature[32,33,49] and those currently annotated in the ClinVar database[50]. We include in addition 31 variants that are annotated as benign or of "uncertain significance" in the ClinVar database as controls[50]. A total of 75 AAT variants are missense variants except for three variants that generate a truncated protein (Y62*, E281* and Null Hong Kong (NHK)) that severely disrupt function (Fig. 1a). This collection of variants includes the most common missense variants in AAT in terms of allele frequency (AF) that have been reported in the general population (gnomAD database[51]) including M1 (V237A; 22% AF), M2 (R125H; 15.6% AF), M3 (E400D; 27% AF), S (E288V; 2.3% AF), Z (E366K; 1.1% AF) and others that have >0.1% AF. The residues impacted by the variant collection are spread across the entire AAT sequence (Fig. 1a, b, lower panel, brown balls), thus providing molecular fiduciary markers[1] that enables us to probe the sequence-based folding and function space defining the ensemble of AAT structures found in the extant population in response to genetic variation.

To characterize the impact of a variant on the status of the AAT protein fold responsible for its monomeric or polymeric states based on its folding, stability and trafficking itinerary through the secretory pathway (Fig. 1c), we utilize two conformation-specific antibodies, a monomer-specific antibody 16f8 (Supplementary Fig. 1a) and a polymer-specific antibody 2C1[52]. These antibodies were used in enzyme-linked immunosorbent assays (ELISA) (Supplementary Fig. 1a, b) to measure the level of AAT monomer or polymer in intracellular and extracellular environments (Fig. 1c) using high-throughput formats (see "Methods"). To measure the inhibitory activity of secreted AAT variants to neutrophil elastase (NE), its natural substrate in the lung, we used a sensitive fluorogenic NE substrate (Z-AAAA)2Rh110[53] (Supplementary Fig. 1b) (see "Methods"). A high level of NE inhibitory activity prevents the digestion of the fluorogenic NE substrate resulting in a low fluorescence signal (Supplementary Fig. 1c). Combined, use of the conformation-specific ELISA and NE inhibitory activity assays allows us to assess the different folding and functional states of AAT variants across its biological itinerary from the ER in the liver to secretion out of the cell for delivery to downstream tissues (Fig. 1c).

### Defining the folding and functional properties of AAT variants
To generate a comprehensive understanding of the functional phenotypic properties of AAT in response to its physiologic secretion itinerary in the liver (Fig. 1c), WT and 75 variants were transfected in a hepatocyte-derived cell line Huh7.5 with endogenous AAT silenced by CRISPR/Cas9 (Huh7.5^null)[54–56]. We also transfected these variants in a human bronchial epithelial lung cell line IB3 that has no detectable endogenous AAT[57] to capture epithelial pathways potentially contributing to lung function. The level of intracellular or extracellular AAT variants in either monomer or polymer form, and the NE inhibitory activity of the extracellular (secreted) pool of AAT variants were measured (Fig. 1d, Huh7.5^null cells; Supplementary Fig. 2, IB3 cells). The measured activity values for secreted monomer, intracellular polymer and NE inhibitory activity are highly correlated between the liver cell line Huh7.5 and the lung IB3 cell line (Supplementary Fig. 3a). These results demonstrate that the basic folding and functional features of AAT variants are conserved across different cellular environments.

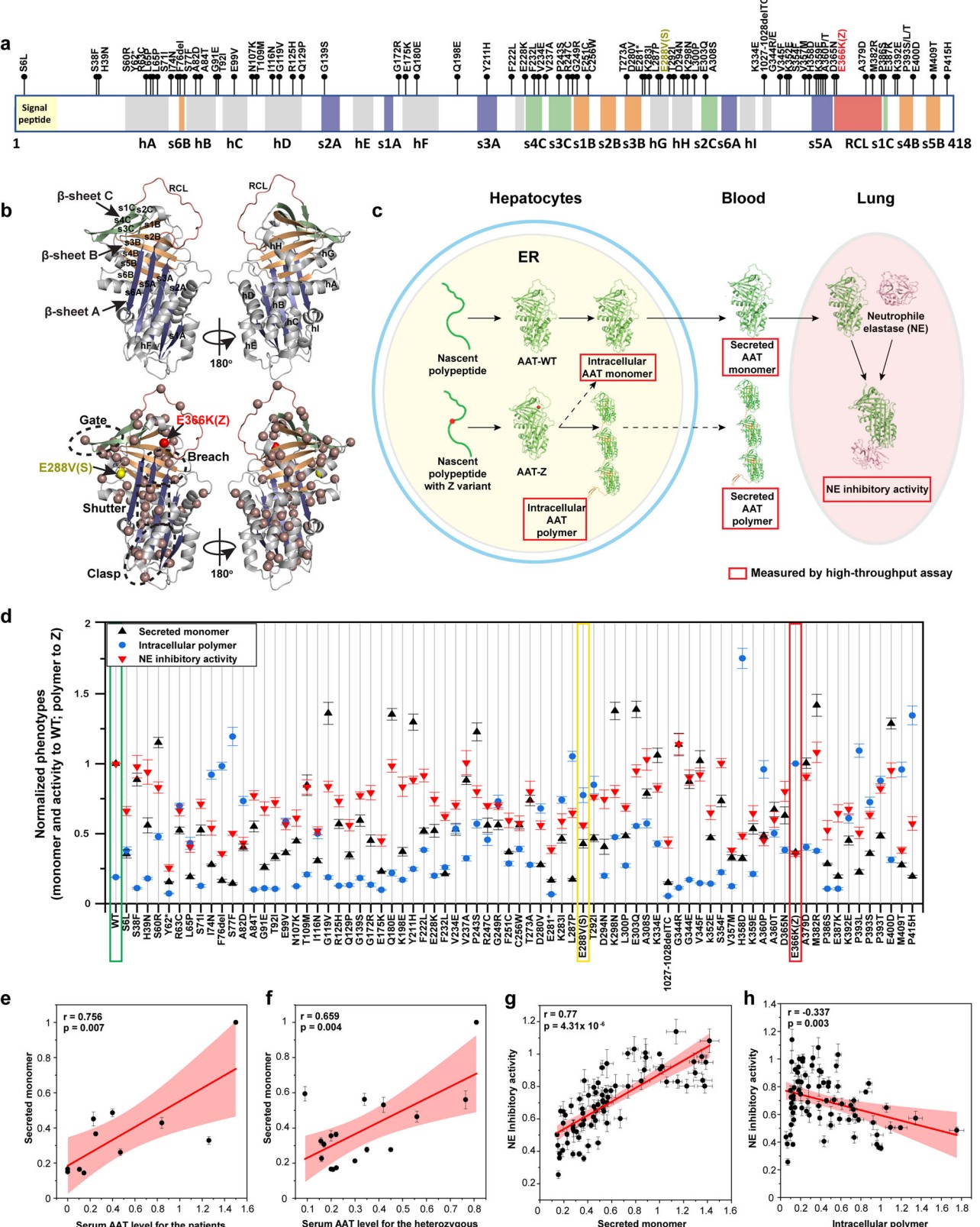

Furthermore, for each variant the intracellular level of monomer and polymer conformations are highly correlated with extracellular monomer and polymer (Supplementary Fig. 3b). These results demonstrate that extracellular levels of monomer and polymer generally reflect the intracellular monomer and polymer, respectively. Given the similarity of the conformational features associated with each AAT variant between the intracellular and extracellular space, we use the measured levels of intracellular polymer, secreted monomer and the NE inhibitory activity from Huh7.5$^{null}$ cells henceforth to understand how, on a residue-by-residue basis, the folding of AAT is coupled to function to drive different disease states found in the AATD population.

**Fig. 1 | Quantification of secreted monomer, intracellular polymer, and NE inhibitory activity for AAT variants. a** 75 AAT variants from the worldwide population investigated in this study are distributed at different structural elements across the entire AAT polypeptide sequence. Eight α-helices (hA-hI) are indicated by gray blocks. Three β-sheets comprising s1-6A (sheet A), s1-6B (sheet B), and s1-4C (sheet C) are highlighted by blue, orange, and green, respectively. The reaction central loop (RCL) is illustrated by the pink block. Z (E366K) and S (E288V) variants are labeled in red and yellow, respectively. **b** Distribution of the variants in the 3D structure of AAT (PDB: 3NE4[67]). The alpha carbon of the variant residues are shown as brown balls. The gate, breach, shutter, and clasp regions[141] are indicated. β-sheets A, B, and C are highlighted. **c** WT AAT is synthesized in the endoplasmic reticulum (ER) of hepatocytes and secreted as a monomer into circulation for delivery to lung to perform its Neutrophil elastase (NE) inhibitory activity. AAT variants such as Z (E366K) lead to intracellular polymerization that reduces AAT secretion and function. We have developed high-throughput assays (indicated by red squares) to measure the intracellular monomer and polymer, secreted

monomer and polymer, and NE inhibitory activity for each of the AAT variants. **d** The levels of secreted monomer, intracellular polymer and NE inhibitory activity for WT-AAT and 75 AAT variants transfected in Huh7.5[null] cells are shown (see "Methods"). The secreted monomer and NE inhibitory activity are normalized to WT values. The intracellular polymer is normalized to AAT-Z value. **e, f** Correlation between measured secreted monomer levels and reported serum AAT levels in AATD patients who are homozygous with the indicated variant genotype (**e**), or from heterozygous patients who share the common Z allele (**f**). **g, h** Correlation between the NE inhibitory activity of AAT variants and the secreted monomer (**g**) or intracellular polymer levels (**h**). The Pearson's r values and the corresponding $P$ values (one-way ANOVA) for the presented correlations are indicated. 95% confidence intervals of the correlation are indicated by light red region. Data is presented as means ± SD. Sample size $n = 3$ biologically independent measurements for secreted monomer, NE inhibitory activity and intracellular polymer of each variant. The sample size for the patients with reported AAT serum levels for different genotypes was indicated in the Source Data file.

We first compared AAT-WT to AAT-Z. As expected, AAT-Z shows low monomer secretion (~37% of WT) and low NE inhibitory activity (~36% of WT), while AAT-WT shows low intracellular polymer (~20%) when compared with AAT-Z (Fig. 1d, red rectangle vs green rectangle). These results are consistent with the severe liver and lung-disease phenotypes for patients with homozygous Z triggered by the combined gain of toxicity in the liver due to accumulation of AAT-Z intracellular polymer and loss-of-function in the lung due to AAT-Z deficiency[30]. In contrast, the S allele (E288V) shows a median level of secreted monomer, intracellular polymer and NE inhibitory activity (Fig. 1d, yellow rectangle vs green rectangle), consistent with the mild disease phenotypes of patients with homozygous S allele. We found that our cell-based measurements of secreted monomer levels are significantly correlated with reported AAT serum levels from patients with different variants, indicating the utility of our cell-based assay to capture the clinical features of disease in the patient population (Fig. 1e, f). Moreover, the secreted monomer levels strongly correlate with their NE inhibitory activity (Fig. 1g, Pearson's $r = 0.77$, $P = 4.3 \times 10^{-6}$), indicating that the monomer conformation we measure corresponds to the functional form of AAT. In contrast, the intracellular polymer levels of the variants are only moderately anti-correlated with the extracellular NE inhibitory activity values (Fig. 1h, Pearson's $r = -0.34$). While variants with high intracellular polymer (Fig. 1h, polymer ≥1 (levels equal or above of AAT-Z)) generally show extracellular NE inhibitory activity lower than the mean of measured NE inhibitory activity of all variants (~73% of WT activity), variants with low intracellular polymer (Fig. 1h, polymer <0.2) have very diverse NE inhibitory activities ranging from ~26% of WT to 114% of WT (Fig. 1h), suggesting a low polymerization propensity does not necessarily confer high functional activity.

We categorize AAT variants into three classes based on their NE inhibitory activity. We choose a 75% cutoff threshold for loss-of-function given that of the 40 AAT variants that have been reported to cause significant clinical disease phenotypes, 33 of them (>80%) have NE inhibitory activity lower than 75% of WT activity and the mean of the predicted NE inhibitory activity for all variants is around 73%. We designate class I variants are loss-of-function defined as a NE inhibitory activity <75% of WT AAT; class II contains the function-impaired variants with NE inhibitory activity between 75% and 90% of WT AAT; class III variants show NE inhibitory activity comparable to WT AAT (>90% of WT activity) (Table 1). For the variants that have lower NE inhibitory than WT AAT, we further divide them into variants with high polymer (>50% of AAT-Z polymer) and low polymer (<50% of AAT-Z polymer). In the class I category comprising the function loss variants, more than a half do not show high polymer, suggesting that polymerization is not the major driving force for the functional deficiency of these variants. These results suggest a more complicated set of relationships between folding, stability and the functional properties of AAT variants

contribute to changes in the protein fold responsible for disease that need to be managed by proteostasis in the AATD population.

## GRP94 ATPase modulation rescues AATD phenotypes for AAT-Z
To test the potential role of GRP94 on the phenotypic diversity imposed by AAT variants, we used PU-WS13 (Fig. 2a), a GRP94-specific inhibitor that blocks the ATPase activity of GRP94[22–24]. The IC50 of PU-

**Table 1 | Classification of AAT variants based on the level of NE inhibitory activity and intracellular polymer**

| Class I | | Class II | | Class III |
|---|---|---|---|---|
| **Activity loss variant**[a] | | **Activity impaired variant**[b] | | **Full activity variant**[c] |
| **High polymer variant**[d] | **Low polymer variant**[e] | **High polymer variant**[d] | **Low polymer variant**[e] | |
| R63C | S6L | P243S | S60R | WT |
| I74N | Y62* | T292I | A84T | S38F |
| F76del | L65P | P393T | T109M | H39N |
| S77F | S71I | | G119V | Q180E |
| A82D | G91E | | G139S | F222L |
| E99V | T92I | | G172R | V237A |
| I116N | N107K | | K198E | E303Q |
| V234E | R125H | | Y211H | A308S |
| G249R | Q129P | | T273A | G344R |
| D280V | E175K | | K298N | G344E |
| K283I | E228K | | K334E | V345F |
| L287P | F232L | | D365N | S354F |
| E288V(S) | R247C | | | A379D |
| H358D | F251C | | | M382R |
| A360P | C256W | | | E400D |
| A360T | E281* | | | |
| E366K(Z) | D294N | | | |
| K392E | L300P | | | |
| P393L | 1027-1028delTC | | | |
| P393S | K352E | | | |
| M409T | V357M | | | |
| P415H | K359E | | | |
| | P386S | | | |
| | E387K | | | |

*Stop codon.
[a]Class I variants with NE inhibitory activity <75% of WT.
[b]Class II variants with NE inhibitory activity between 75 and 90% of WT.
[c]Class III variants with NE inhibitory above 90% of WT.
[d]Variants with intracellular polymer above 50% of AAT-Z.
[e]Variants with intracellular polymer below 50% of AAT-Z.

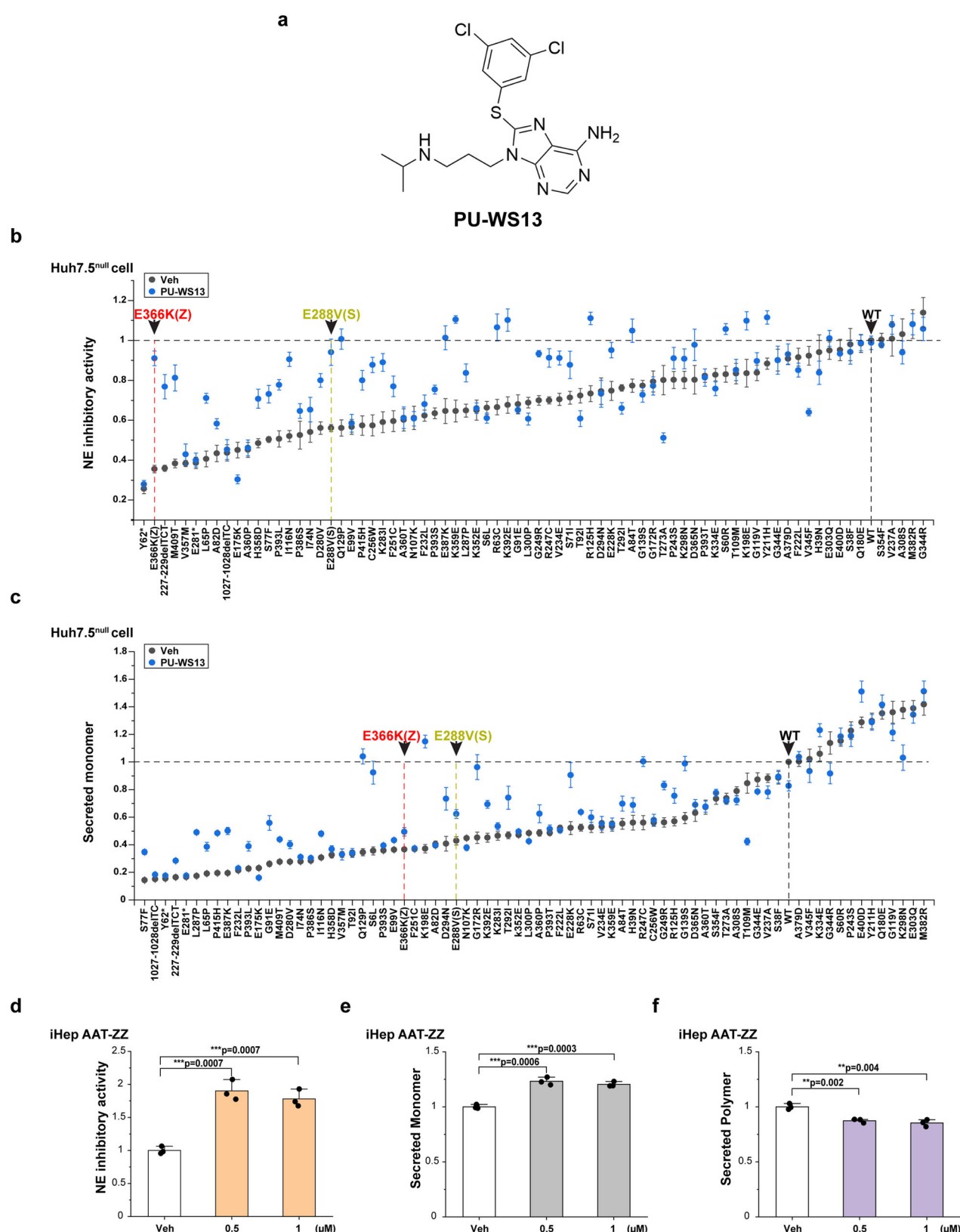

**a** PU-WS13

**b** Huh7.5null cell

**c** Huh7.5null cell

**d** iHep AAT-ZZ

**e** iHep AAT-ZZ

**f** iHep AAT-ZZ

WS13 for GRP94 inhibition is 0.22 μM, while the IC50 of PU-WS13 for inhibiting HSP90α, HSP90β, and TRAP1 are 27.3 μM, 41.8 μM, and 7.3 μM, respectively[23]. We tested PU-WS13 in different doses in Huh7.5 cells (see below Supplementary Fig. 10a–e) revealing 1 μM PU-WS13 treatment achieved the maximum correction for Z-AAT NE activity, monomer secretion, and reducing polymer burden. In total, 1 μM

PU-WS13 treatment should block most GRP94 ATPase activity while avoiding potential off-target effect on other Hsp90 family members.

We first tested the effect of PU-WS13 on the NE inhibitory activity using the fluorogenic NE substrate (Z-AAAA)2Rh110[53] for wild-type (WT) AAT (AAT-WT) and an additional 75 AAT variants transiently transfected in a liver-derived Huh7.5 cell line in which AAT gene has

**Fig. 2 | GRP94 ATPase inhibitor PU-WS13 rescues the NE inhibitory activity, and monomer secretion of AAT variants. a** The chemical structure of PU-WS13. **b, c** The responses of AAT variants to PU-WS13 (1 µM) in Huh7.5[null] cell for NE inhibitory activity (**b**) and secreted monomer (**c**). The variants were ordered by basal condition values from lowest value to highest value. WT, Z allele and S allele are labeled and highlighted by arrows. Data are presented as mean ± SD, *n* = 3 biologically independent measurements. **d–f** Human iPSC-derived AAT-ZZ hepatocytes (iHep AAT-ZZ) (iHepZZ) were treated in the presence or absence of PU-WS13 at 0.5 µM and 1 µM for 24 h. The NE inhibitory activity of secreted AAT-Z proteins from iHep AAT-ZZ cell was measured using fluorogenic substrate of NE (**d**). Secreted AAT-Z monomer was measured by ELISA using monomer-specific antibody 16F8 (**e**). The secreted polymer was measured by ELISA using polymer-specific antibody 2C1 (**f**). Data are presented as mean ± SD, *n* = 3 biologically independent measurements. Student's t test, two tailed. *$P < 0.05$; **$P < 0.01$; ***$P < 0.001$; N.S., $P > 0.05$.

been knocked-out (Huh7.5[null])[54,55]. Intriguingly, we found that PU-WS13 treatment significantly increased the NE inhibitory activity for 37 of all 76 (~49%) tested variants (Fig. 2b). For the 37 variants with improved NE inhibitory activity in response to PU-WS13 treatment, 30 (~81%) of them have less than 75% of AAT-WT NE inhibitory activity in the vehicle control state (Fig. 2b), indicating a potential key role for GRP94 in managing the function of defective AAT variants.

As the NE inhibitory activity strongly correlates the monomer secretion in the basal state in the absence of drug treatment, we tested the impact of PU-WS13 on monomer secretion for 76 AAT variants using the conformation-dependent ELISA assay. PU-WS13 treatment significantly increases the monomer secretion for 40 of all 76 (~53%) tested variants (Fig. 2c). For the 40 variants with improved monomer secretion in response to PU-WS13 treatment, 38 (~95%) of them have monomer secretion levels less than 75% WT in the vehicle basal state (Fig. 2c). These results suggest that GRP94 impacts the features of AAT fold features that contribute to both its monomer secretion and NE inhibitory activity, including the prominent AAT-Z and AAT-S alleles contributing to clinical disease (Fig. 2b, c).

To validate the correction effect of PU-WS13 on the NE inhibitory activity and monomer secretion of AAT variants (Fig. 2b, c) with an endogenous promotor in a more physiological environment (Supplementary Fig. 4a), we tested the impact of PU-WS13 in hepatic lineages prepared from iPSCs derived from a homozygous AAT-ZZ patient expressing AAT under its endogenous promoter (iHepZZ) (Fig. 2d–f and Supplementary Fig. 4b, c). Compared to secretion levels observed iHepZZ cells under the endogenous promoter (Supplementary Fig. 4a), transient transfection of AAT-Z in our experimental cell Huh7.5[null] cell line generates an ~1.2-fold increase of monomer secretion and ~1.3-fold of polymer secretion (Supplementary Fig. 4a). Treatment of iHepZZ cells by PU-WS13 at either 0.5 µM or 1 µM substantially increases both the total secretion of AAT (Supplementary Fig. 4b, c; 1.5-fold and 1.7-fold, respectively) and the NE inhibitory activity (Fig. 2d, ~1.7-fold), suggesting that the functional improvement by PU-WS13 treatment is applicable for AAT variants expressed under the endogenous promotor found in the patient-derived cellular environment. Notably, PU-WS13 increased NE inhibitory activity of AAT-ZZ in iHepZZ cells (Fig. 2d) is a value close to that of AAT-MZ individuals who are largely healthy[38,58]. Whereas we found that the secretion of AAT-Z monomer is significantly improved (Fig. 2e), the polymer secretion was decreased by PU-WS13 treatment when measured by ELISA using polymer-specific antibody 2C1 (Fig. 2f)[52]. Consistent with these observations, native gel analysis of the secreted AAT-Z from Huh7.5[null] cells shows that the monomer fraction is increased while the polymer fraction is decreased by PU-WS13 treatment (Supplementary Fig. 4d–f). The improvement of total secretion and NE inhibitory activity of AAT-Z, the increase of both intracellular and secreted AAT-Z monomer pools, and the decrease of both intracellular and secreted AAT-Z polymer by PU-WS13 treatment are prevented by reducing of GRP94 expression using siRNA (Supplementary Fig. 5), suggesting that the effect of PU-WS13 is specific for GRP94. These results indicate that the inhibition of GRP94 ATPase activity by PU-WS13 not only increases the level of AAT-Z secretion, but also improves the quality of secreted AAT-Z by decreasing the secretion of polymer, resulting in an improvement of NE inhibitory activity.

## Using SCV to define functional rescue across the AAT sequence

While many AAT variants are rescued by PU-WS13 when expressed in Huh7.5[null] cells or iHepZZ cell, it is a variable response reflecting the differential impact of each residue on AAT folding and function managed by GRP94 (Fig. 2b, c). Compared to the basal state (Supplementary Fig. 6a; Pearson's $r = 0.77$, $P = 4.3 \times 10^{16}$), PU-WS13 treatment significantly reduces the correlation between the level of secreted monomer and NE inhibitory activity (Supplementary Fig. 6b, Pearson's $r = 0.59$, $P = 2.6 \times 10^{-8}$). This result suggests GRP94 ATPase inhibition by PU-WS13 can, surprisingly, differentially manage the relationship between folding and functional properties of AAT on a global residue-by-residue basis.

To begin to understand the mechanism by which GRP94 modulation corrects the AAT fold on a residue-by-residue basis at atomic resolution, we applied our Gaussian process (GP)-spatial covariance (SCV) (GP-SCV) principled machine learning approach through variation spatial profiling (VSP)[1,17–21]. GP-SCV principled relationships generated through VSP is based on a statistical paradigm used to find value in complex physical landscapes (see "Methods"). VSP integrates sparse genotype information found in the worldwide population to predict the role of each amino acid residue contributing to a phenotype in the context of all other residues in the polypeptide sequence with defined uncertainty[1,34]. Given the impact of secreted monomer and NE inhibitory activity on lung disease, we plotted the normalized residue positions for all the measured missense variants (Fig. 3a, x axis, full-length polypeptide chain noted "1") relative to their secreted monomer levels (Fig. 3a, y axis) to understand the impact of GRP94 on NE inhibitory activity for the entire protein sequence (Fig. 3a, z axis, color scale) in the absence (Fig. 3a, left panel) or presence of PU-WS13 (Fig. 3a, right panel).

GP-SCV principled relationships first compute the separation distances between any pairwise combinations of the input variants based on their variant residue positions along the polypeptide sequence and the y axis feature, in this case secreted monomer levels (Fig. 3a, black lines). VSP subsequently generates the associated variance with NE inhibitory activity for each pairwise comparison (see "Methods"). These values are modeled by a variogram to illustrate how the NE inhibitory activity changes according to the differences of monomer secretion in relationship to its position in the polypeptide sequence (Fig. 3b). While we found a strong correlation between monomer secretion and the NE inhibitory activity across the entire polypeptide in the native basal state suggesting that a strong integration of folding and function features are required for secretion (Fig. 3b and Supplementary Fig. 6c, black line), PU-WS13 treatment dramatically shortened the distance range for covariance to 0.11 (i.e., covering ~46 residues) (Fig. 3b, blue line). These results suggest that reduced GRP94 ATPase activity relaxes the stringency for many sequence regions of the fold to allow a more diverse collection of monomer-activity covarying relationships to be secreted from ER[1,17,34]. Furthermore, the plateau value of the variogram representing the global variance of NE inhibitory activity is reduced by PU-WS13 treatment (Fig. 3b and Supplementary Fig. 6c), consistent with a general restoration of NE inhibitory activity to WT level for most AAT variants (Fig. 2b).

Variogram modeling of SCV relationships between known variants provides a rigorous platform for GP to generate a multi-dimensional

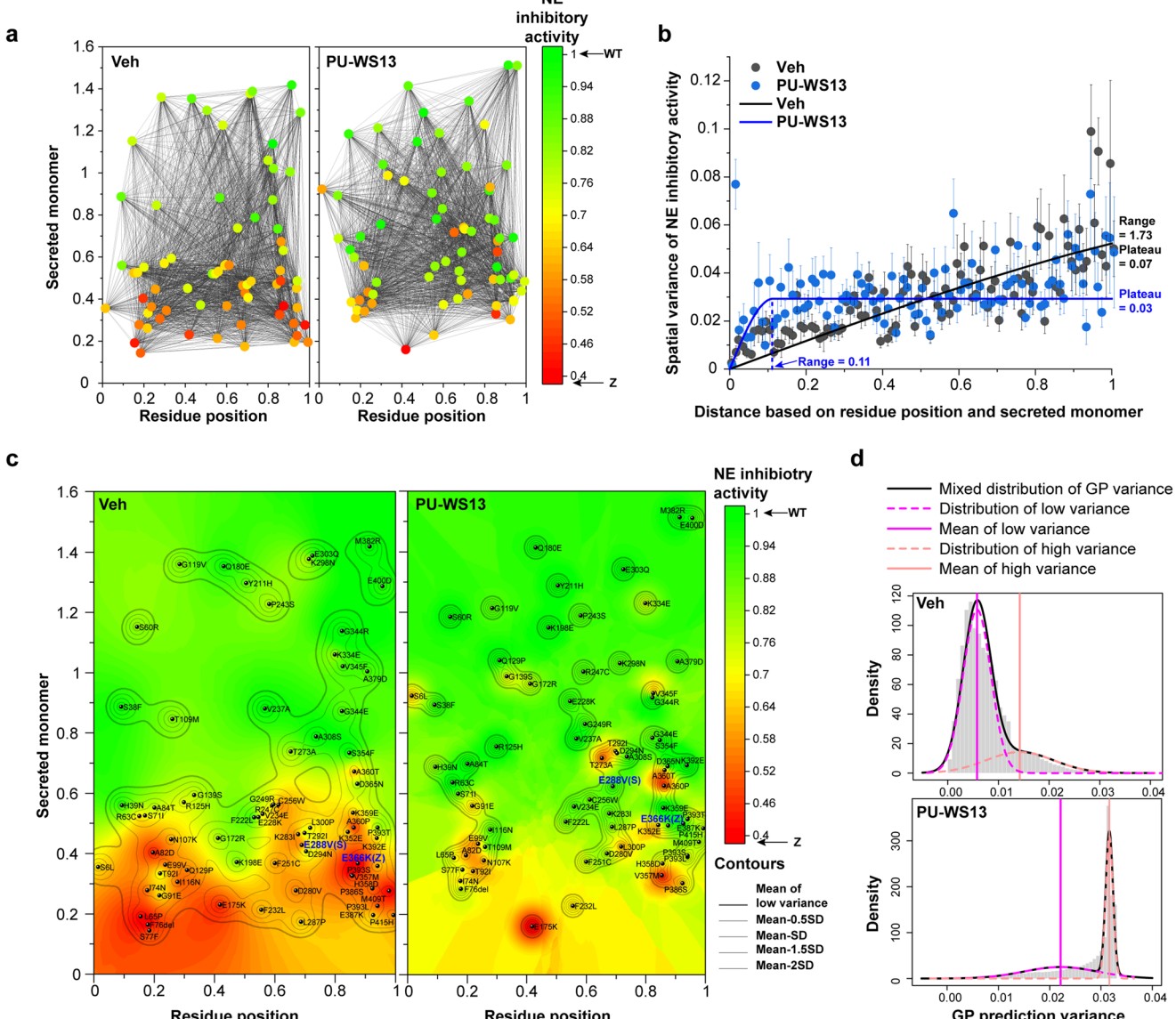

**Fig. 3 | GP-based phenotype landscapes in response to GRP94 ATPase inhibition. a** AAT variants are organized by their variant residue position (*x* axis) normalized by the full-length polypeptide sequence, secreted monomer (*y* axis) and NE inhibitory activity (*z* axis, color scale) in the absence (left panel) or presence (right panel) of PU-WS13. **b** All possible pairwise combinations of variants are analyzed (illustrated as black lines in (**a**)). The relationships between the spatial variance of NE inhibitory activity and the distance values defined by variant residue positions and monomer secretion are modeled by molecular variograms in the absence (black dots and line) or presence (blue dots and line) of PU-WS13 (left panel). Data are presented as mean ± SEM, *n* of pairwise combinations based on biologically independent measurements of variants is indicated in the Source Data file. The correlation distance range and plateau value of each variogram are indicated.

**c** Phenotype landscapes generated by GP-based VSP approach linking secreted monomer (*y* axis) and NE inhibitory activity (*z* axis, color scale) across the entire AAT polypeptide residue positions (*x* axis) in the absence (left panel) or presence (right panel) of PU-WS13. **d** A two-component Gaussian mixture model to separate the low vs high GP-generated variance for each prediction in the absence or presence of PU-WS13. The density of the separated distributions for low variance (magenta dash line) and high variance (pink dash line) are shown. The mixed distribution is illustrated as a black curve. The mean of the low variance distribution (magenta line) and high variance distribution (pink line) are indicated. The mean of the distribution of low GP variance and the standard deviation (SD) below the mean are illustrated as contours in the phenotype landscapes (**c**) to indicate high-confidence predictions.

'phenotype landscape' to map the GP-SCV principled relationships relating AAT sequence position to monomer secretion to NE inhibitory activity for every residue in the AAT polypeptide chain where input *x*- and *y*-coordinate values are used to predict the *z*-coordinate value for all residues[1] (Fig. 3c) (see "Methods"). As a probabilistic platform, GP-SCV based analysis also generates an associated uncertainty (i.e., a GP-based variance) for each prediction output based on variant input (Fig. 3d). A two-component Gaussian mixture model is used to fit the distribution of the associated uncertainty to separate the high-confidence vs low-confidence predictions (Fig. 3d). The mean of the

distribution for the low uncertainty is illustrated in the landscape as the outermost bold contour and values lower than the mean are illustrated by the inner contours using the standard deviation of the distribution (Fig. 3c). The regions with uncertainty below the mean capture high-confidence predictions found in the phenotype landscape for every residue across the full sequence, generally encompassing 95% of all residues[1,17–21] attesting to the strength of the approach for assigning value across the entire landscape (Fig. 3c). The variogram parameters required for phenotype landscape build can also be modeled using a Bayesian approach[59–62] to provide a

comprehensive quantification of uncertainty in the modeling (Supplementary Fig. 11). We found that the predictions and associated uncertainties generated by a more extensive Bayesian approach were similar to GP-SCV predictions using our best-fit variogram parameters (Supplementary Fig. 11a, b; Pearson's $r = 0.99$ for prediction; Pearson's $r = 0.98$ for uncertainty) (Supplementary Fig. 11), indicating the value of a computationally tractable approach to address disease management based on SCV principled relationships.

The phenotype landscape predicting the NE inhibitory activity in the basal state (Fig. 3c, left panel, vehicle control) achieves significant prediction accuracy based on the leave-one-out cross-validation (Pearson's $r = 0.75$, $P = 5 \times 10^{-14}$). Furthermore, the correlations between the predictions and experimental values are strong for the phenotype landscape at residues 1–139 (Pearson's $r = 0.78$, $P = 1.7 \times 10^{-5}$), 140–278 (Pearson's $r = 0.7$, $P = 0.003$) and 279–418 (Pearson's $r = 0.73$, $P = 9 \times 10^{-7}$), illustrating the high value of the platform even for select regions distributed across the entire polypeptide chain. Strikingly, PU-WS13 treatment dramatically changes the NE inhibitory activity phenotype landscape, shifting large areas found in the AAT basal state landscape with defective NE inhibitory activity (Fig. 3c; left panel, red–orange–yellow) to normal activity (Fig. 3c, right panel, green), though there remain regions in the AAT polypeptide sequence that are insensitive to correction by PU-WS13 (Fig. 3c, right panel, red–orange). In the basal state landscape, most of the variants are well connected mechanistically with one another, triggering functional deficiency through high-confidence SCV relationships as illustrated by the broad contour lines (Fig. 3c, left panel, contours). PU-WS13 treatment largely shrinks and isolates these high-confidence regions (Fig. 3c, right panel, contours; Fig. 3d) and decreases the prediction accuracy (Pearson's $r = 0.45$, $P = 9 \times 10^{-5}$), consistent with the short spatial correlation range observed in the variogram in the presence of PU-WS13 (Fig. 3b, blue line). These results suggest that reduced GRP94 ATPase activity in response to PU-WS13 alters the residue–residue SCV relationships leading to dysfunction in the AAT basal state to a more flexible folding state resulting in an increase in both monomer secretion and the NE inhibitory activity. Improving the general plasticity of SCV relationships could be a common mechanism by which proteostasis handles mutational load to restore function, in this case through the management of GRP94 ATPase activity that modulates GRP94 cycling. These results suggest a very dynamic role for GRP94 binding of client when transitioning the polypeptide chain set-point from unfolded to more functional folded states. We next mapped this conversion on a residue-by-residue basis by viewing each residue as part of a collective of states reflected in their SCV relationships to every residue in the polypeptide chain.

## Phenotypic mapping GRP94 ATPase managed to function for every residue in AAT sequence

The phenotype landscapes report a range of relationships between NE inhibitory activity and secreted monomer for each residue in the AAT sequence (Fig. 4a, a 3D visualization of Fig. 3c). Taking residue 366 where the AAT-Z variant is located as an example (Fig. 4a, gray slice), we can plot the predicted relationships between NE inhibitory activity and secreted monomer for this residue as a curve in the presence (Fig. 4b, blue curve) or absence (Fig. 4b, black curve) of PU-WS13. The error bars represent the standard deviation of each prediction generated by GP-based variance to indicate the prediction confidence, and the high-confidence predictions defined by the Gaussian mixture modeling (Fig. 3d) are highlighted by darker error bars (Fig. 4b). The GP-based predictions in response to PU-WS13 on residue 366 (Fig. 4b, blue curve) are strikingly different from the basal state (Fig. 4b, black curve) in the range of monomer secretion from ~30% to ~60% of WT (Fig. 4b, $0.3 < x < 0.6$), where the NE inhibitory activity is greatly improved by PU-WS13 treatment, as shown by AAT-Z variant (Fig. 4b, arrows). In contrast, GRP94 ATPase inhibition does not generate a

significant impact in the range of WT-like monomer secretion (Fig. 4b, $x > 0.6$), suggesting that GRP94 mainly manages the variants associated with misfolding and defective trafficking.

To understand the contribution of each residue in the correction of the pathogenic AAT fold by PU-WS13, we used inverse variance weighting (IVW) to average the predicted NE inhibitory activity in the context of monomer secretion across the entire landscape for each residue[21]. IVW allows us to assign the likelihood of function for a given residue position for the entire sequence that will be disrupted by variants with diverse chemical properties[21], emphasizing that it is the probabilistic value of function derived from the collective of all variants that can be used to generate the SCV maps that rigorously address the role a given residue in the AAT wild-type protein sequence fold function. As expected, this function can be disrupted to different degrees by the different chemical properties of a particular variant residue found in the population impacting severity of disease. By using the reciprocal of prediction uncertainty (i.e., GP-based variance) as weight, IVW prioritizes the prediction with high confidence (i.e., low uncertainty/GP-based variance) over the prediction with low confidence (i.e., high uncertainty/GP-based variance) during the averaging (Fig. 4b). Through GP analyses (Fig. 4a, b), the IVW averaged value reports the most likely role of each residue contributing to the NE inhibitory activity based on SCV relationships. We also applied IVW to the phenotype landscape generated by Bayesian analysis, which yielded almost identical residue-based NE inhibitory activity for each residue across the AAT polypeptide (Supplementary Fig. 11f, g; Pearson's $r = 0.98$), suggesting the robustness of this approach to estimating the most likely contribution of each residue to the function. The mean values for each residue are plotted as linear barcodes from the N-terminal to C-terminal of AAT polypeptide chain reflecting the NE inhibitory activity at the basal state (Fig. 4c, upper barcode (DMSO vehicle)) and after the treatment of PU-WS13 (Fig. 4c, middle barcode), as well as the delta (Δ) value between them (Fig. 4c, lower barcode). At the basal state with the DMSO vehicle control (Fig. 4c, upper barcode), the IVW barcode map defines three sequence regions at the N-terminal (N1), middle (M2) and carboxyl-terminal (C3) regions along the AAT polypeptide that are clustered with variants disrupting the NE inhibitory activity (Fig. 4c; labeled brackets, N1, M2, and C3) reflecting potential hotspots for managing protein misfolding.

These unanticipated results based on our sequence analysis and published structural efforts to date[63–65] suggest that AAT has evolved both short- and long-range cooperative interactions to manage NE inhibitory activity that could be sensitive to proteostasis management. Strikingly, all these regions are largely rescued by PU-WS13 treatment (Fig. 4c). Specifically, hA-s6B-hB and hD in the N1 region, a region around the S variant in M2 and most of C3 region, are highly rescued by PU-WS13 (Fig. 4c). These results suggest that GRP94, from a SCV perspective, can act at specific hotspots that involve different features of the fold protein spanning the entire protein sequence, possibly driven by regional fold energetics[66], that as a collective globally respond to not just variation in the population but to proteostasis management to improve activity. We next explored how this conversion occurs.

## GRP94 ATPase activity can manage long-range residue interactions to correct AAT activity

To understand the structural mechanism(s) of correction by PU-WS13 treatment, we mapped the NE inhibitory activity predicted by GP analysis for each residue to the AAT 3D structure (PDB:3NE4)[67] to generate a "functional-structure" before (Fig. 4d) or after PU-WS13 treatment (Fig. 4e), as well as the structure representing the delta (Δ) value in response to PU-WS13 treatment (Fig. 4f). Interestingly, though N1, M2, and C3 sequence regions that contribute the NE inhibitory activity at the basal state are separated in the primary sequence of the polypeptide (Fig. 4c, upper barcode), they interact with each other through the β-sheet B (Fig. 4c, s6B strand for N1, s1-3B strand for M2

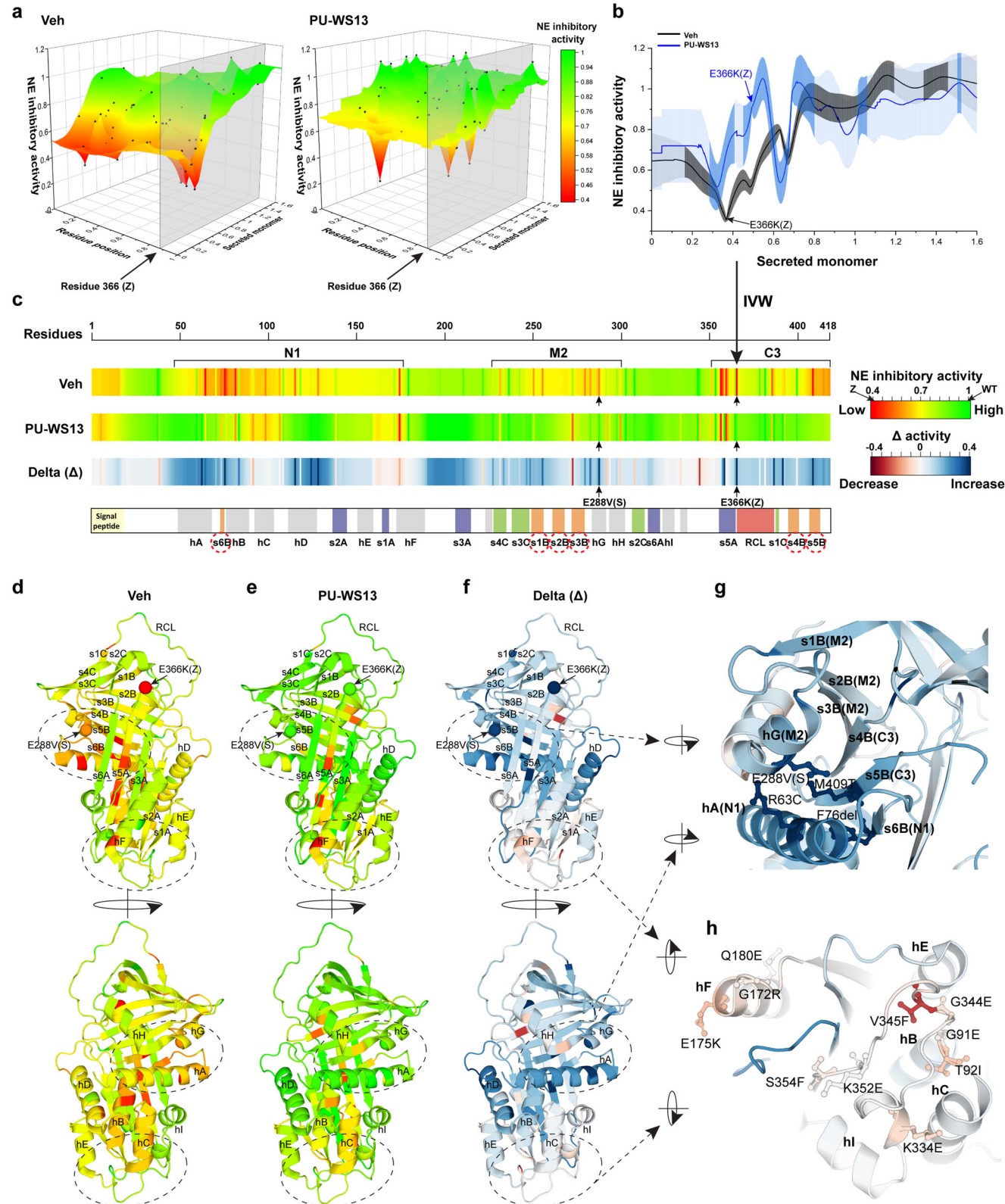

and s4-5B strand for C3, red dotted circles; Fig. 4d). The integration of each of the core SCV regions through β-sheet B indicates that the long-range interactions are crucial to evolve the function-structure relationships defining the AAT architecture (Supplementary Fig. 6d–h).

PU-WS13 administration generates corrections on many residues in the AAT structure to restore WT-like NE inhibitory activity to the AAT fold (Fig. 4e vs Fig. 4d, red–orange–yellow to green). Intriguingly, we discovered a core region that is highly responsive to PU-WS13

management, which encompasses residue interactions between hA-s6B from region N1, hG from M2 and s5B from region C3 (Fig. 4f, g). Notably, the highly responsive region to PU-WS13 treatment also involves the interactions between the N-terminal β-strand s6B and C-terminal β-strand s5B that are critical for assembling β-sheet group B (Fig. 4g). These results reveal that the integrity of β-sheet group B is critical for GRP94 managed NE inhibitory activity. These observations are consistent with the view that GRP94, as a member of the Hsp90

**Fig. 4 | Residue-by-residue responses of NE inhibitory activity to PU-WS13. a** To project a 3D view of the phenotype landscapes, the predicted NE inhibitory activity is shown on the *z* axis of the landscape in the absence (left panel) and presence (right panel) of PU-WS13. The predicted values for residue 366 where AAT-Z variant is located are highlighted by gray slices. **b** The data highlighted by the gray slice in (**a**) for residue 366 is plotted in the absence (black line), and the presence (blue line) of PU-WS13 is presented as mean ± SD. The dark-gray or dark-blue error bars indicate the high-confidence predictions defined by the two-component Gaussian mixture modeling (Fig. 3d). These high-confidence values are used in the inverse variance weighting (IVW) (see "Methods") to compute the most likely NE inhibitory activity for each residue. This procedure is performed for each residue comprising

the AAT polypeptide sequence. **c** The IVW computed values for each residue are plotted from N-terminal to C-terminal to report residue-based NE inhibitory activity barcode in the absence (upper barcode) or presence (middle barcode) of PU-WS13. The delta (Δ) values between them are presented as the lower barcode. The regions harboring residues that are <75% WT NE inhibitory activity under basal conditions are labeled as N1, M2 and C3. The secondary structure elements of the AAT sequence are indicated in the bottom panel. **d–h** Mapping the residue-based NE inhibitory activity in the absence (**d**) and presence (**e**) of PU-WS13, and their delta (Δ) values (**f**) to AAT 3D structures (PDB:3NE4). The structural region highly responsive to PU-WS13 is zoomed in (**g**), while the non-responsive structural region is zoomed in (**h**).

family, is generally involved in later stages of protein folding or in post-translational folding[14,15,68,69]. Here, reduced ATPase activity may generate a more favorable kinetic/thermodynamic window for the N-terminal to C-terminal cooperative folding events disrupted by variants in the population, thereby significantly improving NE inhibitory activity correction (Fig. 4f, g). These results provide a rationale for the dynamic regulation of Hsp90 family members by co-chaperones[27–29] that could ensure coordination of folding with variation in sequence in the population to achieve an optimal folded state for function. Interestingly, non-responding or negatively impacted residues are clustered in the "bottom" of the AAT structure, comprising hC, s1A-hF, and the loop following hI to the beginning of s5A (Fig. 4c, lower barcode, Fig. 4f, h). These residue–residue relationships could potentially manage post-ER features involved in NE inhibitory activity. Overall, GP-principled SCV relationships provide a residue-by-residue roadmap of how GRP94 ATPase inhibition can differentially manage the AAT fold in response to variation to reshape/reinvent sequence-to-function-to-structure relationships.

### Residue-by-residue-based correction of AAT monomer secretion

As GRP94 ATPase inhibition changes NE inhibitory activity and its relationship to monomer secretion for a number of variants, we wanted to understand the residue–residue basis for the correction of monomer secretion. For this purpose, we used NE inhibitory activity as the input *y* axis feature to predict as output the role of each residue in monomer secretion as the *z* axis feature of the phenotype landscape (Supplementary Fig. 7). PU-WS13 treatment reduced both the correlation range and the plateau value in the variogram modeling and decreased the leave-one-out cross-validation (from Pearson's $r = 0.65$, $P = 5 \times 10^{-10}$ to Pearson's $r = 0.32$, $P = 0.007$) (Supplementary Fig. 7b–d). These results are consistent with the interpretation that PU-WS13 treatment decreases the more rigid coupling between NE inhibitory activity and monomer secretion for AAT variants compared to the basal state without PU-WS13 treatment (Supplementary Fig. 6a, b).

To generate a residue-by-residue description of monomer secretion response to PU-WS13 treatment, we averaged the monomer secretion phenotype landscapes for each residue (Supplementary Fig. 7d) using IVW to generate residue-based monomer secretion barcodes (Fig. 5a). These mean values were mapped onto the AAT structure (Fig. 5b–f). We observed an increase of monomer secretion in the core cluster we identified above as a high response cluster for improving NE inhibitory activity (Figs. 4g and 5e). This core region comprises the interactions between hA-s6B (N1), hG (M2) and s5B (C3), suggesting that the high response of NE inhibitory activity in this region to PU-WS13 treatment (Fig. 4g) is at least partially due to the improvement of monomer secretion (Fig. 5e). However, the improvement of monomer secretion is much weaker (Fig. 5e) when compared to the NE inhibitory activity response (Fig. 4g), suggesting that there are additional effects generated by PU-WS13 treatment that increases the NE inhibitory activity independent of the level of monomer secretion. Indeed, we found that the top responding residues and structural regions for monomer secretion enhancement by PU-WS31

treatment (Fig. 5a, lower barcode; Fig. 5d) are in different sequence positions compared to those rescuing NE inhibitory activity (Fig. 4c, lower barcode; Fig. 4f). For example, the highly responsive region s2A-hE-s1A leading to improved monomer secretion (Fig. 5a, lower barcode; Fig. 5d, f) does not result in a reciprocal improvement of NE inhibitory activity (Fig. 4c, lower barcode; Fig. 4f). These results further suggest that GRP94 ATPase inhibition increases the tolerance of the ER for AAT folding in a manner that can differentially impact the monomer secretion and the NE inhibitory activity on a residue-by-residue basis. We next explored the impact of these changes on intracellular polymer accumulation- the hallmark of liver disease[32,70–72].

### Managing the N-terminal to C-terminal covariance folding of AAT to reduce polymer load

Given that the AAT-ZZ variant results in both intracellular and secreted polymer pools that can be corrected by PU-WS13 treatment (Fig. 2f and Supplementary Figs. 4d–f and 5f, h), we wanted to understand the molecular basis for these events across the entire AAT sequence. Therefore, we measured both the intracellular (Fig. 6a) and secreted polymer (Supplementary Fig. 8a) for all AAT variants in the absence or presence of PU-WS13 using an ELISA assay based on the polymer-specific antibody 2C1 (see ref. 52). Among all measured 76 variants, 27 of them have intracellular polymer level at levels that are at least 50% of the severe intracellular AAT-Z polymer pool (Fig. 6a). In all, 13 of the 27 variants with high intracellular polymer (~48%) showed significant reduction in the intracellular polymer pool in the presence of PU-WS13 (Fig. 6a). Similarly, among all measured 76 variants, 20 of them have secreted polymer above 50% of the secreted AAT-Z polymer level (Supplementary Fig. 8a). PU-WS13 treatment significantly reduced the secreted polymer for 11 of the 20 variants with high secreted polymer (~55%) (Supplementary Fig. 8a). These results suggest that PU-WS13 has a broad impact on the AAT fold to decrease the polymer pools for around a half of the polymerogenic AAT variants, a metric highly relevant to monitoring liver disease in the patient population[32,70–72].

Consistent with the clinical observations that serum polymer level can be used as a biomarker for liver disease[71,72], the measured secreted polymer is strongly correlated with intracellular polymer for all the variants in both vehicle DMSO control (Supplementary Fig. 8b, left panel; Pearson's $r = 0.92$, $P = 4.3 \times 10^{-31}$) and PU-WS13 treatment condition (Supplementary Fig. 8b, right panel; Pearson's $r = 0.85$, $P = 2 \times 10^{-22}$). This result indicates that the reduction of secreted polymer for AAT variants by PU-WS13 treatment is likely due to the reduced level of the intracellular polymer as suggested before[73–75].

To generate a residue-by-residue response map to PU-WS13 treatment for the intracellular polymer triggering liver disease, we built phenotype landscapes using secreted polymer as the y-coordinate to predict the intracellular polymer (z-coordinate) for every residue comprising the AAT polypeptide sequence (Supplementary Fig. 8c–f). We then used IVW to generate the residue-based barcodes (Fig. 6b) and related functional structure of AAT (Fig. 6c–f) to annotate the impact of PU-WS13 on intracellular AAT polymer pools. We also mapped the response of intracellular polymer pools to PU-WS13 to the more native AAT-Z polymer structure recently

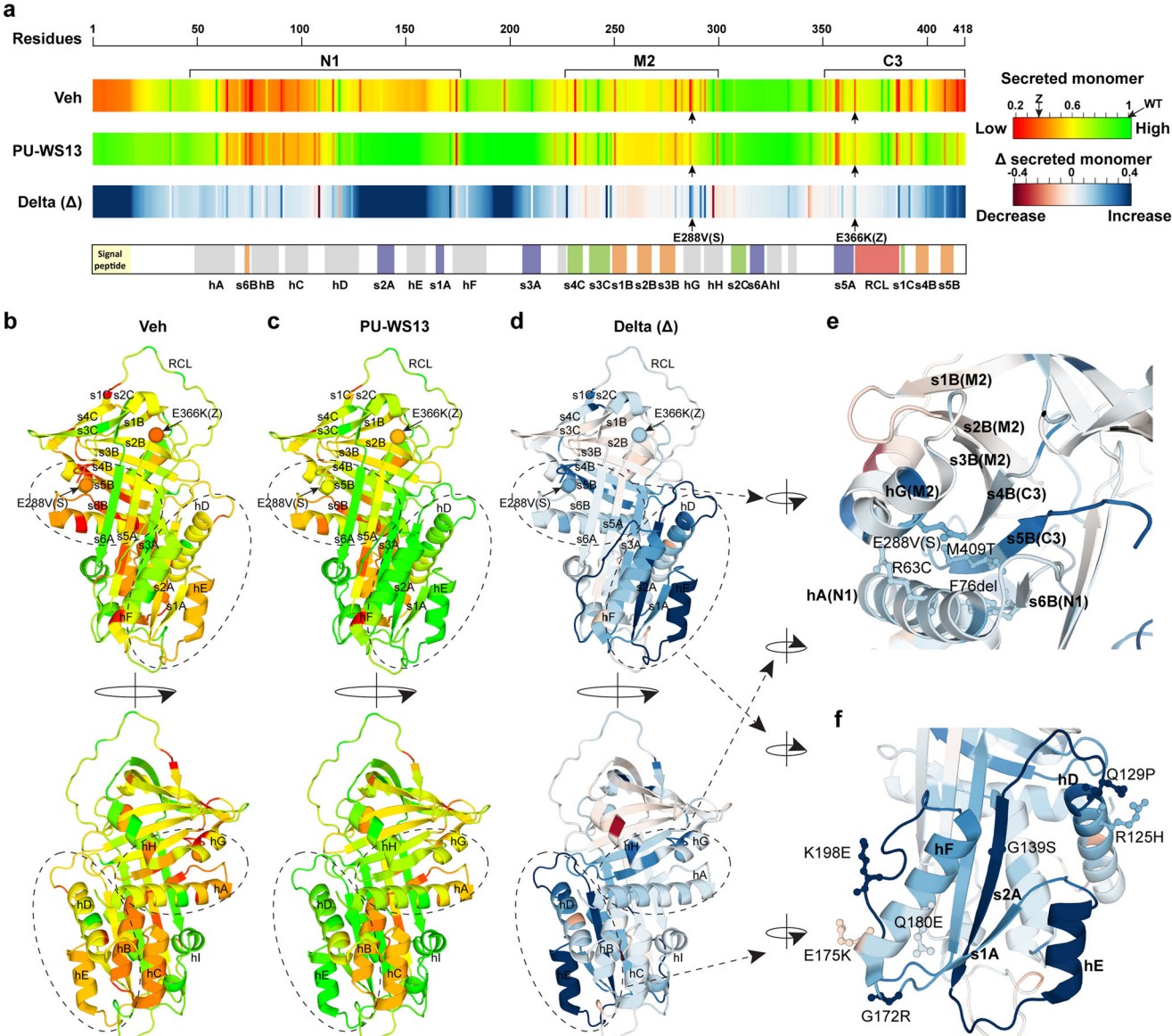

**Fig. 5 | Residue-by-residue responses of monomer secretion to PU-WS13.**
**a** Residue-based monomer secretion barcodes derived from the phenotype landscapes through IVW in the absence (upper barcode) or presence (middle barcode) of PU-WS13. The delta (Δ) values between them are presented as the lower barcode. **b–f** Mapping the residue-based monomer secretion in the absence (**b**) and

presence (**c**) of PU-WS13, and their delta (Δ) values (**d**) to AAT 3D structure. The region that is highly corrected on NE inhibitory activity illustrated in Fig. 4e is zoomed in for comparison to the PU-WS13 impact on monomer secretion (**e**). The top responding structure region for monomer secretion is zoomed in (**f**).

obtained using protein isolated from a ZZ-homozygous patient liver that revealed a "C-terminal polymerization" cryo-EM model[63,64]. In the 'C-terminal polymerization' model, not only the RCL inserts as s4A in β-sheet A by "loop-sheet" insertion mechanism but also s4B and s5B insert into the β-sheet B of another molecule[63,64] (Fig. 6g-i). At the AAT basal state (Fig. 6c, g), the GP-based covariance prediction for polymer assembly is consistent with the C-terminal polymerization model (Fig. 6g) that highlights the critical role of assembly of β-sheet B in the polymer formation triggered by variation in the AATD population (Fig. 6c, g).

Strikingly, the residue interactions between s6B from the N1 region and s5B from the C3 region are clearly the highly responsive regions to PU-WS13 leading to significant polymer reduction (Fig. 6b, lower barcode; 6e, f, i). In contrast, the polymerogenic residues found in s5A that are critical for the "loop-sheet" insertion polymerization mechanism do not or only weakly respond to PU-WS13 (Fig. 6e, i). These results reveal that GRP94 ATPase inhibition by PU-WS13

primarily manages the N-terminal to C-terminal cooperative folding driving β-sheet B assembly to reduce the polymer accumulation in the liver-derived cells[63,64]. Thus, by profiling the residue-by-residue response to GRP94 ATPase inhibition using GP-principled SCV relationships, we have uncovered an unanticipated mechanism used by the GRP94 chaperone to manage the polymerization state of AAT fold triggering aggregation disease.

**Tracing the global management of function on a residue-by-residue basis by GRP94 ATPase activity**

To generate a global view of how inhibition of GRP94 ATPase activity can differentially manage each residue of AAT fold for polymer accumulation in the ER, monomer secretion, and NE inhibitory activity in the hepatocyte in response to human variation, we overlaid the residue-based Δ value for each of these features in response to PU-WS13 treatment (Fig. 7a, gray (monomer), magenta (NE activity), cyan dots (intracellular polymer)). We observed a general improvement for

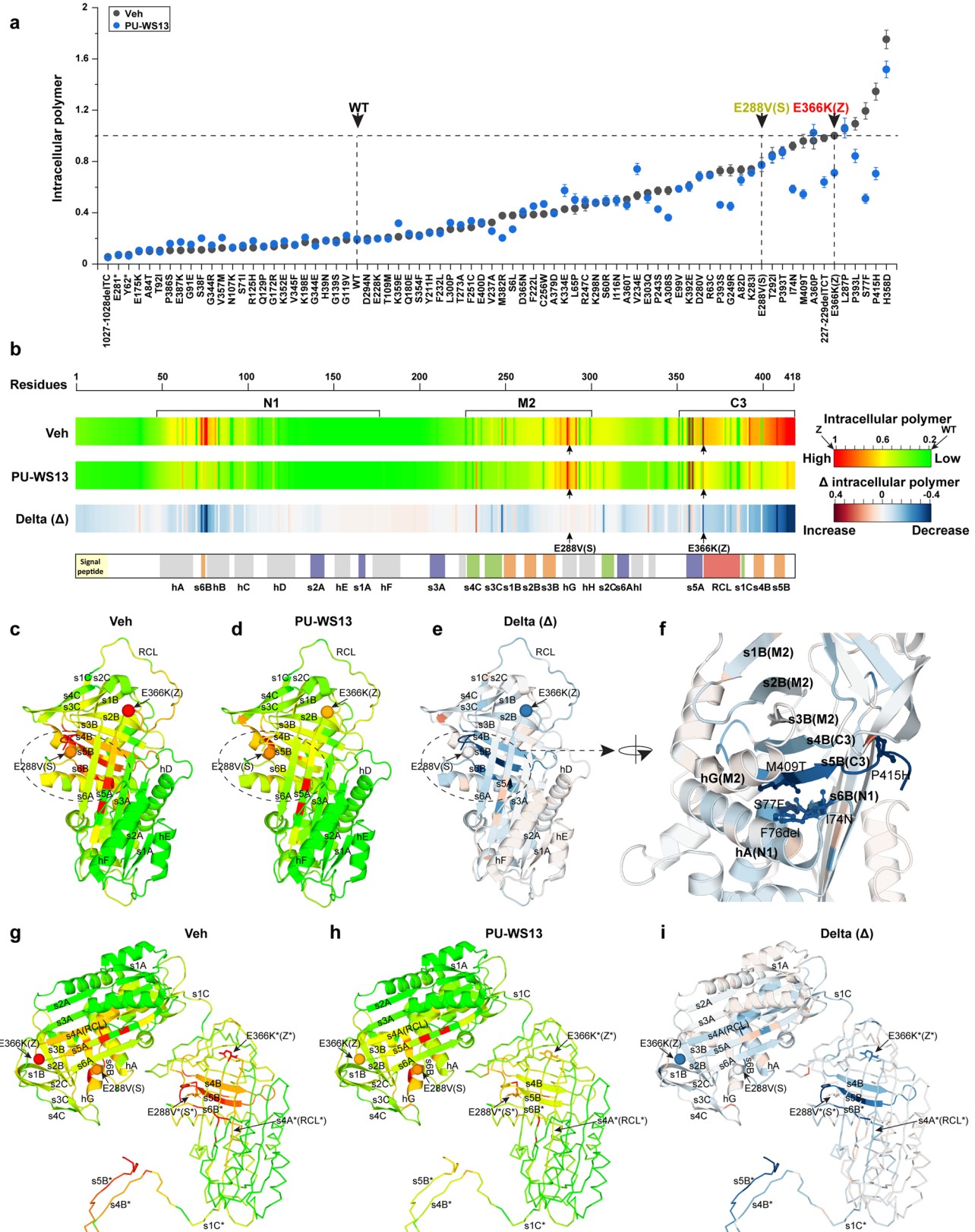

**Fig. 6 | Residue-by-residue responses of intracellular polymer to PU-WS13.**
**a** The responses of AAT variants to PU-WS13 in Huh7.5[null] cell for intracellular polymer. Data are presented as mean ± SD, $n = 3$ biologically independent measurements. **b** Residue-based intracellular polymer barcodes derived from the phenotype landscapes through IVW in the absence (upper barcode) or presence (middle barcode) of PU-WS13. The delta (Δ) values between them are presented as the lower barcode. **c–f** Mapping the residue-based intracellular polymer in the absence (**c**) and presence (**d**) of PU-WS13, and their delta (Δ) values (**e**) to AAT monomer structure. The highly responding region to PU-WS13 for the intracellular polymer is zoomed in (**f**). **g–i** Mapping the residue-based intracellular polymer in the absence (**g**) and presence (**h**) of PU-WS13, and their delta (Δ) values (**i**) to AAT polymer structure (PDB: 3T1P).

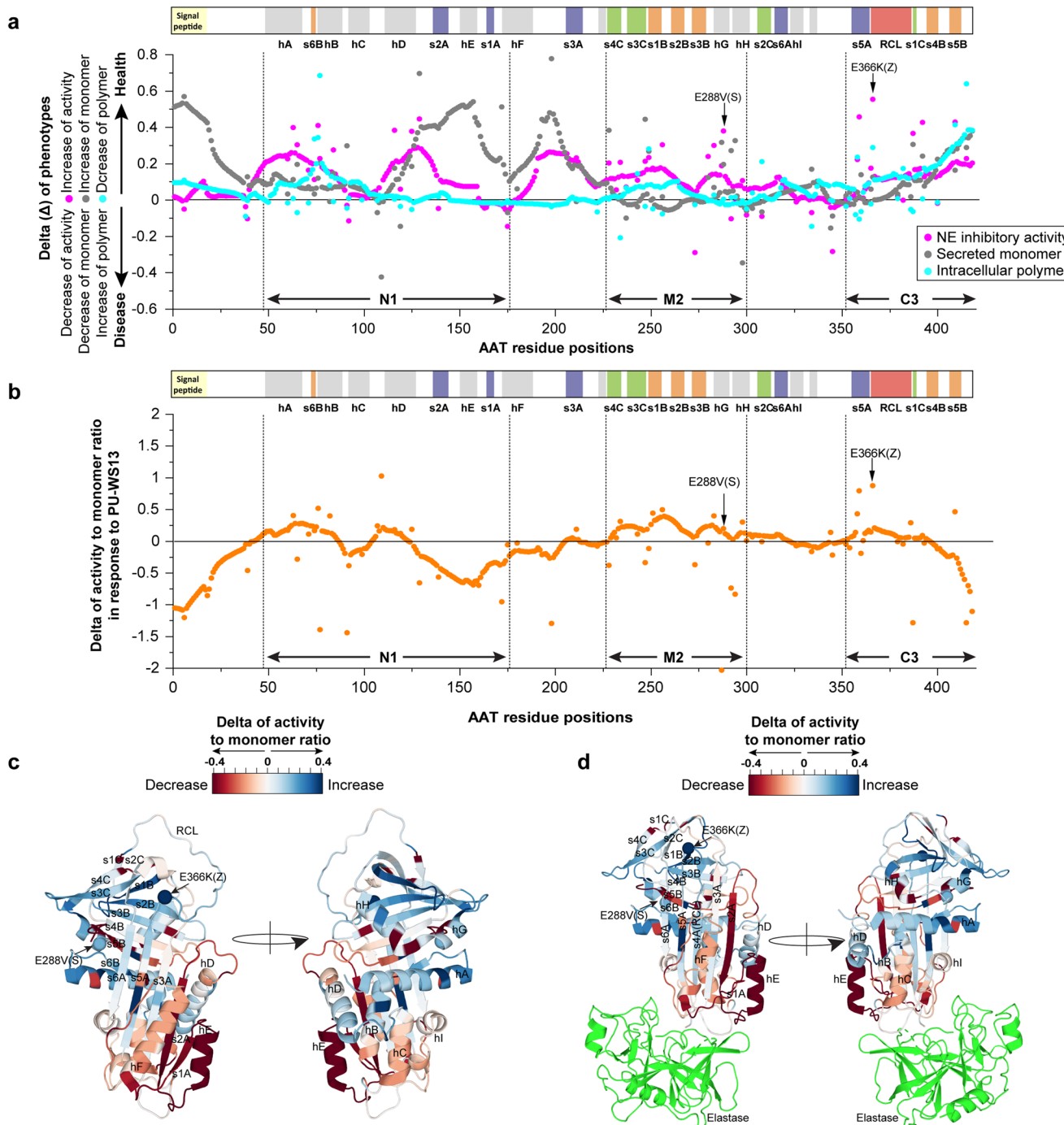

**Fig. 7 | Differential responses of different phenotypes for each residue of AAT to GRP94 ATPase inhibition. a** Overlay of the delta value between the DMSO vehicle and PU-WS13 treatment for the residue-based NE inhibitory activity (magenta), monomer secretion (gray), and intracellular polymer level (cyan). Positive delta values indicated as improvement of NE inhibitory activity, an increase of monomer secretion and decrease of intracellular polymer. N1, M2, and C3 sequence regions, which are in rich of variants leading to defective NE inhibitory activity at basal state, are labeled. **b** Residue-by-residue activity to monomer ratio in response to PU-WS13. The NE inhibitory activity and secreted monomer ratio values are computed from both vehicle and PU-WS13 states for each residue. The delta value of the activity-to-monomer ratio in response to PU-WS13 for each residue is plotted. **c, d** Mapping the delta (Δ) of activity to monomer ratio on AAT monomeric structure (**c**) and the complex structure of AAT-elastase (PDB: 2D26) (**d**).

all the three features on the N-terminal half of the N1 region encompassing hA-s6B-hB and the C3 region housing the s4B-s5B strands, suggesting overall a critical role of GRP94 managed N-terminal to C-terminal cooperative folding events to assemble β-sheet B to prevent polymer accumulation, leading to enhanced monomer secretion and NE inhibitory activity through manipulation of its ATPase activity (Fig. 7a). Interestingly, we also observed sequence regions that show

improvement of monomer secretion and NE inhibitory activity without the correction of intracellular AAT polymer. For example, there is no change in the intracellular polymer pool for the sequence region spanning the C-terminal half of the N1 region to the beginning of the M2 region (Fig. 7a), residues that do not contribute to polymer formation in the basal state but instead contribute to monomer secretion and NE inhibitory activity. Notably, PU-WS13 treatment

does generate a general and large improvement in both monomer secretion and NE inhibitory activity for selective sequence fragments in this region (Fig. 7a). These results further emphasize that the rescue of export from ER and restoration of activity by reduced GRP94 ATPase activity can be separated from its impact on polymer. A second example of the stabilizing activity of GRP94 in response PU-WS13 is found in the C-terminal half of M2 region (Fig. 7a). This region contains hG-hH where the AAT "S allele" is located (Fig. 7a). PU-WS13 treatment only weakly impacts its lower intracellular polymer pool when compared to the Z-variant pool, but does increase both monomer secretion and NE inhibitory activity (Fig. 7a). These results further emphasize that GRP94 ATPase inhibition by PU-WS13 can differentially manage the fold properties for specific residue regions to improve monomer secretion and NE inhibitory function in a manner that can be either dependent or independent on the AAT polymer status.

To better quantitate these residue-specific differential responses, we calculated the delta (Δ) of the ratio between the NE inhibitory activity and secreted AAT monomer (referred to hereafter simply as "activity-to-monomer ratio") for each residue in response to PU-WS13 treatment (Fig. 7b) and mapped these values onto AAT monomer structure (Fig. 7c), or the AAT-NE complex structure (PDB: 2D26)[76] (Fig. 7d). Interestingly, we found that the M2 region shows a general increase of the activity-to-monomer ratio (Fig. 7b–d). M2 includes s4C-s3C that comprises the previously suggested "gate" area[77,78], and s1B-s3B-s3B-hG-hH that forms "latch" interactions with the "gate"[77], both of which have been suggested to regulate the loop-sheet mechanism required for extracellular AAT function[77–79]. Furthermore, s5A-RCL in C3 region that forms the loop-sheet insertion as s5A-s4A in the functional cycle of AAT with NE (Fig. 7d) also shows an increased activity-to-monomer ratio (Fig. 7b–d). These results suggest that for the region associated with the loop-sheet insertion mechanism, GRP94 ATPase inhibition potentially generates a more flexible AAT fold to improve the monomer 'specific activity' in its reaction with NE. In contrast to the improvement of the activity-to-monomer ratio, there are residues that show reduced responses (Fig. 7b–d). Interestingly, they are mostly clustered at the binding interface between AAT and NE (Fig. 7d, red residues), suggesting that the increased extracellular AAT monomer pool generated in response to PU-WS13 for these residues reflects a disrupted interaction with NE.

In summary, GP-principled SCV relationships generated in the absence or presence of PU-WS13 reveals at a residue-by-residue level an unanticipated pliable state of the protein fold and its sequence-to-function-to-structure relationships responsive to GRP94 ATPase activity—providing a plausible mechanistic view of Hsp90 family ATPase activity in shaping the protein fold in response to variation in the population.

## Discussion

To evolve protein form and function requiring dynamic and metastable structural changes, a general computational framework is needed for understanding the balance in sequence-to-function-to-structure relationships dictating the genome to proteome transformation. A key feature in managing protein folding and function in response to human variation and the environment is proteostasis. In the case of AATD, past efforts focused on developing small molecules or peptides that block the aggregation of AAT-Z by interfering with the loop-sheet insertion mechanism responsible for NE inhibitory activity[40,80–84]. While preventing polymer formation in the liver, they were of limited utility for lung disease in AATD given that the loop-sheet insertion is essential for AAT function. Another strategy to reduce AAT-Z polymer load in the ER was to activate the autophagy pathway to remove polymer burden[85–87], an approach that does not address the critical loss-of-function in the lung responsible for the vast majority of pathologies observed in the clinic.

By profiling the genetic and phenotypic diversity associated with AATD through GP-SCV principled relationships in response to the ATPase inhibitor PU-WS13, we uncovered unexpected and precision residue–residue coupled relationships that simultaneously correct the defects in polymer formation, monomer folding, secretion, and NE inhibitory activity of most pathogenic AAT variants. This detailed phenotypic information cannot be captured in a conservation score derived from sequence alignment across different species (Supplementary Fig. 9), suggesting that GP-SCV principled relationships capture complex evolutionary information driving function revealed by the natural process of genetic diversity in the human population. Whereas past studies have focused on chemical kinetic stabilization of the fold to prevent diseases of misfolding, including transthyretin[88,89] and immunoglobulin light chain amyloidosis[90], Gaucher's[91,92] and cystic fibrosis[21,93], GP-SCV provides a fresh approach to understand more completely the complex features of cooperative folding and fold architecture to develop insights into challenging sequence-to-function-to-structure relationships that could prevent misfolding while maintaining activity.

The improvement in NE inhibitory activity in response to PU-WS13 is largely correlated with improved monomer secretion along with decreasing both intracellular and secreted polymer for most AAT variants, including AAT-Z variant, shifting the folding balance from disease to a healthier state of the fold (Fig. 8). Specifically, a reduction of GRP94 ATPase activity revealed its role in managing the N-terminal (s6B) to C-terminal (s5B) cooperative folding of AAT essential to assemble β-sheet B contributing to NE inhibitory activity. This conclusion is supported by the 'C-terminal polymerization' model based on the cryo-EM structure study of AAT-Z polymer from patient liver[63], which indicates an important role of the assembly or disassembly of β-sheet B in polymerization of AAT variants leading to aggregation in the ER[63]. Furthermore, the interaction region between s6B and s5B is distant to the reaction loop (RCL), therefore, unlike the loop-sheet mechanism, managing s6B-s5B interactions would not be expected to impact the extracellular NE inhibitory activity, and may even increase the monomer-specific activity as suggested by PU-WS13 treatment on AAT-Z and other variants (Fig. 8).

It is interesting that at a concentration of 1 μM PU-WS13 used in all the assays (Supplementary Fig. 10a–e), we did not observe up-regulation of BiP[94] (Supplementary Fig. 10f). This is in contrast to siGRP94 which up-regulates the level of BiP in macrophages[95]. These results suggest that GRP94 in the PU-WS13 bound form may be required for the rescue effect, for example, to facilitate the release of AAT variant to ERGIC-53 for secretion (Supplementary Fig. 10g, h). Moreover, whereas inhibition of GRP94 through PU-WS13 has been shown to reduce the stability of HER2 in a tumor-specific manner where siGRP94 mimics the effect of PU-WS13 to destabilize HER2[24], in this study we found that whereas PU-WS13 rescues the folding, secretion, and function of AAT-Z, silencing through siGRP94 does not rescue AAT-Z defects. Thus, SCV relationships capture residue-specific features of GRP94 ATPase function affecting its role in promoting uniquely AAT-Z stability and function. These results suggest that the correction effects we observed are part of a larger programmable system in which GRP94 ATPase activity (and potentially other factors[95]) may be normally reprogrammed by co-chaperones to achieve optimal function in response to variation and the environment as has been observed for the Hsp70 family of chaperone/co-chaperone relationships[89,90,96–99].

The residue-by-residue responses to GRP94 ATPase inhibition is different from that we recently found for ATF6 activators that more globally control the differential activities of multiple Hsp70 and Hsp90 chaperones and their respective co-chaperones in the ER in response to misfolding stress and the environment[34]. For example, in terms of NE inhibitory activity, s6B-s5B interactions only moderately respond to ATF6 activators[34]. In contrast, the "gate" area that is strongly impacted

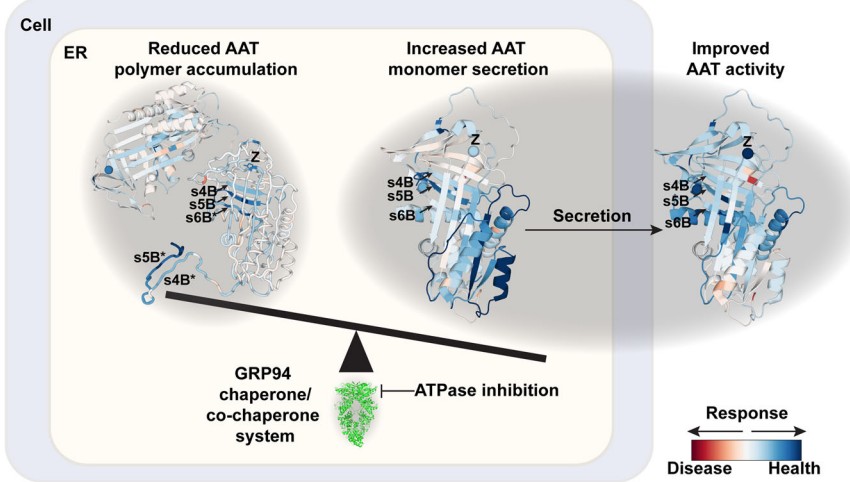

**Fig. 8 | GRP94 manages the N- to C-terminal cooperative folding of AAT to shape the balance between AAT aggregation monomer secretion for function.** The delta (Δ) phenotype structures of NE inhibitory activity (Fig. 4f), monomer secretion (Fig. 5d) and intracellular polymer (Fig. 6i) are presented to illustrate the residue-by-residue responses of AAT (Fig. 7a) to GRP94 ATPase inhibitor PU-WS13. Blue residues indicate the responsive regions for improved AAT activity, increased monomer secretion and reduced polymer accumulation. The highly corrected regions for all the phenotypes by GRP94 ATPase inhibition involve long-range interactions between s6B from N1 region and s4B-s5B from C3 region that are highlighted by arrows. Modulation of GRP94 chaperone/co-chaperone system by ATPase inhibition through PU-WS13 treatment improves the SCV integrity of β-sheet B for AAT variants to rebalance AAT aggregation with function to reduce the intracellular polymer accumulation and increase the monomer secretion and extracellular NE inhibitory activity.

by ATF6 activators[34] is only moderately responsive to GRP94 ATPase inhibitor. These observations highlight the fact that different proteostasis components manage different parts of the protein fold to allow for what we refer to as "functional conversion"—not unlike the PrPc/PrPs transitions leading to prion disease[100,101]. These results raise the possibility that conversion is a common principle in biology[34]. Moreover, whereas Hsp90 family members are generally thought to clamp the clients between the pocket created by the dimer interface in cryo-EM snapshots[102,103], the differing SCV relationships we have now defined for every residue using GP logistics raises the possibility of a potential sequence scanning role for Hsp90 family members for management of the fold trajectory in real time.

We now suggest that understanding GP-SCV principled relationships provides a framework for making use of a naturally occurring programmable system in which the Hsp90 ATPase activity is constantly managed by co-chaperones in response to the environment[16,27–29,69] to achieve optimal function for fitness through variation (see below). As GRP94 has been shown to be involved in the ERAD of AAT variants[45–47], a process that is controlled by ER mannosidase I (ERManI/Man1b1)[104–109], particularly in infants of AATD patient population[108], it is likely that GRP94 establishes a variable level of stringency we have referred to as a plasticity set-point[1,20] that is defined by the activity of the resident co-chaperone regulators in each cell type, and in response to the environment and aging (Fig. 8). Our SCV relationships indicate that GRP94 ATPase inhibition can release this SCV based set-point stringency to allow the secretion of AAT variants with improved monomer-activity relationships- even leading to improved specific activity for some variants despite the presence of the variant residue in the secreted monomer[34]. These results suggest that GP-SCV principled management strategies impacting the balance between folding for function and targeting to degradation, including autophagy pathways[85–87], will be of high value for disease management in the context of a covariant understanding delineating the probabilistic uncertainty inherent in human biology impact by the sequence features contributing to the clinical functional outcomes impacting health along the coupled liver–lung axis.

Our results support the view that the role of proteostasis is but one part of an integrated "quality system (QS)"[20] that is responsive to genetics, development, aging, and the environment to continually reshape sequence-to-function-to-structure relationships driving protein fold dynamics contributing to health and disease[1,17,20,34,110] that is not captured in static structures[21,34]. As it is well established by the pioneering efforts of Lindquist and others[68,111–127] that Hsp90 serves as a capacitor for natural selection and evolution, SCV principled relationships now suggest that Hsp90's role in capacitance management of variation occurs on a residue-by-residue basis- but as a collective. The response of Hsp90 family members to ATPase inhibitors suggests their role in cellular QS[20] is acutely tuned by the use of different co-chaperone partners to facilitate fitness[6,14,15,44,128]. Understanding the residue-based conversion of the protein fold and function[1,17–21,32,34] through an SCV principled standard model[129] could provide a common framework from a machine learning perspective to assess the role of each of the over 2500 proteostasis components in the management of information flow from the genome to the functional proteome[130,131]. Knowledge of how information flow is managed by proteostasis from a covariance principled standard model could be used to implement precision interventions in a broad range of inherited and somatic human diseases including neurodegeneration[4,5,9,10,132,133] and cancer[43,134–136] where management of variation is key to restoration of healthspan (Fig. 8).

## Methods

### Reagents

DMEM, LHC-8 medium, F-12 medium, fetal bovine serum (FBS), penicillin–streptomycin (P/S), and geneticin (G418) were purchased from Invitrogen Life Technologies Corporation (Carlsbad, CA). FuGENE6 Transfection Reagent Kit was purchased from Promega (Madison, WI). Plasmid DNA purification kit was purchased from QIAGEN Inc. (Valencia, CA). PU-WS13 was initially provided by Dr. Gabriela Chiosis (Memorial Sloan Kettering Cancer Center, NY), and then purchased from Cayman Chemical (Ann Arbor, MI). PU-WS13 was stocked at 10 mM in DMSO and stored at −20 °C. The goat anti-human AAT polyclonal antibody 80A was purchase from ICL Inc (Anaheim, CA) (Cat # GCYT-80A, 1:1000 dilution); The mice anti-human monomeric AAT antibody 16f8 were generated in-house (1:2000 dilution); The mice anti-human polymeric AAT antibody 2C1 and other antibodies used in the study were purchased from Hycult biotech (Wayne, PA) (Cat # HM2289, 1:1000 dilution). The ERGIC-53 antibody (Cat #

13364-1-AP, 1:20,000 dilution) and 6x-His Tag Monoclonal Antibody (HIS.H8) (Cat # 50-173-6447, 1:5000 dilution) were purchase from Proteintech Inc. GRP78(BiP) antibody (Cat # ab108615, 1:2000 dilution), GRP94 antibody (Cat # ab238126, 1:1000 dilution) and GAPDH antibody (Cat # ab8245, 1:5000 dilution) were purchased from Abcam (Waltham, MA). Secondary antibody Goat anti-Mouse HRP antibody (Cat # 32230; 1:5000 dilution), Goat anti-Rabbit HRP antibody (Cat# 32260, 1:5000 dilution) and Mouse anti-Goat HRP antibody (Cat#31400, 1:10,000 dilution) were all purchased from Thermo Fisher Science (Waltham, MA). GRP94 siRNA was purchased from Thermo Fisher Scientific (ID:119656). Human neutrophil elastase was purchased from Innovative Research (Novi, MI). NE fluorescence assay substrate 2Rh110 (Z-Ala-Ala-Ala-Ala) was purchased from Cayman Chemical (Ann Arbor, MI). All other chemicals were purchased from Sigma Chemical (St. Louis, MO).

### Cell culture

IB3 cells stably transfected with AAT-Z, which were provided by T. Flotte, University of Massachusetts Medical School, Worcester, MA, were cultured in LHC-8 medium containing 10% (v/v) fetal bovine serum (FBS) and 100 μg/ml penicillin–streptomycin (P/S). Huh7.5 AAT knockout (KO) cells (Huh7.5[null]), which were provided by Mark Brantly, University of Florida College of Medicine, Gainesville, FL., and Huh7 cells with His-tagged AAT-Z stably expressed (provided by T. Flotte, University of Massachusetts Medical School, Worcester, MA), were cultured in the DMEM/F-12 medium containing 10% FBS and primocin (100 μg/ml). Patient AAT-ZZ hiPSC cells were purchased from DefiniGEN (Cambridge, UK). Directed differentiation of hiPSC into hepatic lineage was carried out following the protocol provided by DefiniGEN.

### AAT variant DNA constructs

AAT variant DNA constructs were in pcDNA3.1 (+) plasmid vector generated by Quintara Biosciences (Cambridge, MA). All plasmid sequences were validated by Genewiz, Inc (San Diego, CA)).

### AAT-specific conformation (monomer or polymer) ELISA assays

Huh7.5 AAT[null] cells were cultured in 96-well plates with the density $2 \times 10^4$/well and transfected with AAT variant plasmids at 0.2 μg/well. After 24-h transfection and another 24-h PU-WS13 treatment, cells were washed with PBS and incubated with 100 μl/well FBS-free culture medium. After 3-h incubation, FBS-free culture medium was collected, and cell lysates were harvested. In total, 20 μl medium or cell lysates from each well were added into the capture antibody (goat anti-human AAT polyclonal antibody 80A) pre-coated and pre-blocked (with BSA) 96-well plate for ELISA assay. After overnight incubation at 4 °C, the medium or cells lysate were washed away by PBST. Conformation-specific antibody 16f8 or 2C1 (mouse anti-human) was added into plate and incubated for 2 h. After washing the conformation antibody by PBST, secondary HRP-conjugated goat anti-mouse antibody was incubated in the plate for 2 h. After washing the secondary antibody, TMB reagent was added into each well for 10 min reaction and then stopped by 2 M $H_2SO_4$. Plates were read by BioTek Synergy H1 Hybrid Reader (Santa Clara, CA) at 454 nm for absorbance. The reading values were normalized by total protein levels of each well sample (measured by Bradford assay). Monomer levels of AAT variant were normalized by WT monomers. Polymer levels of AAT variant were normalized by AAT-Z polymers. Commercial purchased AAT proteins were subject to serial dilution and prepared for ELISA standardized curve.

### AAT-NE inhibitory activity assay

Cell culture medium was collected as described above. In all, 20 μl culture medium was added into the capture antibody (goat anti-human AAT polyclonal antibody 80A) pre-coated and pre-block (with BSA)

plate for overnight incubation. Medium was washed away and then the plate was incubated with Human Neutrophil Elastase 5 ng/well for 2 h at 37 °C. Overall, 25 pmol/well Neutrophil Elastase substrate 2RH110 was added into the plate and incubated for another 1.5 h. The plate was read at BioTek Synergy H1 Hybrid Reader (Santa Clara, CA) for excitation 485 nm and emission at 525 nm. The reading values were normalized by total protein levels for each sample. The anti-elastase activity of AAT variant was normalized by WT-AAT activity.

### Immunoblotting

Plasmids of AAT variants were transiently transfected in Huh7.5 AAT knockout cells. After 24-h transfection and another 24-h PU-WS13 treatment, cells were washed by PBS and then switched to FBS-free culture medium for 3-h incubation. Culture medium was harvested. Cells were washed twice with 1× PBS and then lysed with 50 μl/well of 1× cell lysis buffer (50 mM Tris-HCl, 150 mM NaCl, 1% (v/v), Triton X-100, and protease inhibitors at 2 mg/ml) on ice for 30 min. Samples were collected and then centrifuged at 20,817×$g$ at 4 °C for 20 min. The supernatant was collected. The protein concentration levels of culture medium and cell lysate were determined by the Bradford assay (Bio-Rad, Hercules, CA). Culture medium samples or cell lysate samples were resuspended in 1× SDS sample buffer containing β-mercaptoethanol and incubated at 95 °C for 5 min. Samples containing 20 μg of total protein were separated on a 10% (v/v) SDS-PAGE, transferred to nitrocellulose, and immunoblotted with goat anti-human AAT antibody (80 A). Detection was performed using chemiluminescence and the appropriate horseradish peroxidase-conjugated secondary antibodies. GAPDH was used in general as loading control.

### siRNA-mediated silencing

IB3 cells stably transfected by AAT-Z were plated in 12-well tissue culture dishes and grown to 60% confluency. Silencing of GRP94 was performed by transfecting with RNAiMax (Invitrogen) and a final concentration of 50 nM siRNA (Ambion; siRNA targeting HSP90B1 gene) into cells according to the manufacturer's protocol (Invitrogen). After 48 h of RNA transfected, PU-WS13 was added into the culture medium. AAT proteins were harvested as described above and subjected into immunoblotting analysis.

### Immunoprecipitation assay

Huh7 cells, which stably expressed the His-tagged AAT-Z, were seeded and cultured in 12-well plates (with cell population density $1.2 \times 10^5$/well) for 24-h and then treated with PU-WS13 at different concentrations for another 24 h. Cell lysates were collected as described above and diluted in the equilibration buffer. Samples with equal amount total proteins and equal volumes were incubated with HisPur™ Ni-NTA Magnetic Beads from Invitrogen (Carlsbad, CA) for 1 h in an end-over-end rotator. Magnetic beads were incubated and vortexed with wash buffer to remove the unbound proteins in the supernatant. Elution buffer was added and incubated with magnetic beads for 15 min on the rotator. His-tagged Z-AAT proteins in the supernatant were removed and used for immunoblot analysis.

### Variation spatial profiling (VSP) to build phenotype landscape

The VSP analysis[1,17,21,34] was performed using the gstat package (V2.0)[137,138] in R. VSP is based on a geostatistical prediction technique, Kriging[139], which is widely used to model the spatial dependency as a Gaussian process to interpolate the unmeasured value to construct the phenotype landscape for AAT.

Briefly, AAT variants were positioned by their sequence positions in the polypeptide chain on the "$x$" axis coordinate and their impact on a phenotype on the "$y$" axis coordinate to the impact on another phenotype on the "$z$" axis coordinate. Ordinary Kriging[139] was applied to

model $z_{(x, y)}$ as:

$$z_{(x,y)} = \mu_{(x,y)} + W_{(x,y)} + \epsilon_{(x,y)} \qquad (1)$$

Where $\mu_{(x,y)}$ is a constant local mean. $W_{(x,y)}$ is the spatially correlated stochastic part of variation. $\epsilon_{(x,y)}$ is a white noise that captures measurement error and microscale variation. The spatial structure of the data can be examined by variogram analysis. Suppose the $i$th observation at coordinates $(x_i)$ and $(y_i)$ in a dataset consists of a value $z_{(x_i,y_i)}$. And $j$th observation at coordinates $(x_j)$ and $(y_j)$ in a dataset consists of a value $z_{(x_j,y_j)}$. They are separated by a vector $h$.

The distance $d(h)$ between the $i^{th}$ and $j^{th}$ observations is calculated by:

$$d(h) = \sqrt{(x_i - x_j)^2 + (y_i - y_j)^2} \qquad (2)$$

The $\gamma(h)$-variance for a given distance $d(h)$ is defined by:

$$\gamma(h) = \frac{1}{2}(z_{(x_i,y_i)} - z_{(x_j,y_j)})^2 \qquad (3)$$

where $\gamma(h)$-variance is the semivariance (the degree of dissimilarity) of the $z$ value between the two observations, which is also the whole variance of $z$ value for one observation at the given separation distance $h$, referred to as spatial variance here. The distance $(h)$ and spatial variance $(\gamma(h))$ for all the data pairs are generated by the Eqs. (2) and (3). Then, the average values of spatial variance for each distance interval are calculated to plot the averaged spatial variance versus distance. The fitting of variograms were determined using GS+ Version 10 (Gamma Design Software) by both minimizing the residual sum of squares (RSS) and maximizing the leave-one-out cross-validation result (see below). The variogram enables us to compute the spatial covariance (SCV) matrices for any possible separation vector. The SCV at a distance $(h)$ is calculated by $C(h) = C(0) - \gamma(h)$, where $C(0)$ is the covariance at zero distance representing the global variance of the data points under consideration (the plateau of the variogram). The approach aims to generate the prediction that has minimized estimation error (error variance) which is generated according to the expression:

$$\sigma_u^2 = E\left[(z_u^* - z_u)^2\right] = \sum_{i=1}^{n}\sum_{j=1}^{n}\omega_i\omega_j C_{i,j} - 2\sum_{i=1}^{n}\omega_i C_{i,u} + C_{u,u} \qquad (4)$$

where $z_u^*$ is the prediction value while $z_u$ is the true but unknown value, $C_{i,j}$ and $C_{i,u}$ are SCV between data points $i$ and $j$, and data points $i$ and $u$, respectively, and $C_{u,u}$ is the SCV within location $u$. $\omega_i$ is the weight for data point $i$. The SCV is obtained from the above molecular variogram analysis and the weight $(\omega_i)$ solved from Eq. (4) is used for following prediction. To ensure an unbiased result, the sum of weight is set as one:

$$\sum_{i=1}^{n}\omega_i = 1 \qquad (5)$$

Equations (4) and (5) not only solved the set of weights associated with input observations, but also provide the minimized "molecular variance" at location $u$ which can be expressed as:

$$\sigma_u^2 = C_{u,u} - \left(\sum_{i=1}^{n}\omega_i C_{i,u} + \mu\right) \qquad (6)$$

where $C_{u,u}$ is the SCV within location $u$, $\omega_i$ is the weight for data point $i$, and $C_{i,u}$ are SCV between data points $i$ and $u$. $\mu$ is the Lagrange Parameter that is used to convert the constrained minimization problem in Eq. (6) into an unconstrained one. The resulting minimized molecular

variance assessing the prediction uncertainty presents the confidence level of the prediction. With the solved weights, we can calculate the prediction of all unknown values to generate the complete fitness landscape by the equation:

$$z_u^* = \sum_{i=1}^{n}\omega_i z_i \qquad (7)$$

where $z_u^*$ is the prediction value for the unknown data point $u$, $\omega_i$ is the weight for the known data point, and $z_i$ is the measured value for data point $i$. Leave-one-out cross-validation (LOOCV) is used to validate the computational model because of small sample size modeling[140]. In the LOOCV, we remove each data point, one at a time and use the rest of the data points to predict the missing value. We repeat the prediction for all data points and compare the prediction results to the measured value to generate the Pearson's $r$ value and its associated $P$ value (ANOVA test performed in Originpro version 2020b (OriginLab)).

### Defining the high-confidence prediction using probabilistic clustering

Each prediction on the phenotype landscape is associated with a GP-generated variance to indicate the prediction confidence. To separate the high-confidence vs low-confidence predictions in the GP-generated phenotype landscape, we fit a Gaussian mixture model with two components over the GP-generated variance by using the mclust package (V.5.4.10) in R. Gaussian mixture model is a probabilistic clustering tool that not only separates the predictions with low variance vs high variance, but also outputs the probability distributions for them. We use the mean of the distribution for the low variance as a cutoff and define the predictions with a lower variance than the mean of the distribution as high-confidence prediction. The mean and the standard deviation of the distribution for the low variance (i.e., high confidence) are illustrated as contours in the phenotype landscape.

### Inverse variance weighting (IVW) to build the residue-by-residue phenotype barcode

Phenotype landscapes built based on a sparse collection of input variants map the full range of values describing function (based on the $y$- and $z$ axis metrics) for the entire polypeptide sequence on a residue-by-residue basis ($x$ axis). To get an averaged value of predicted phenotype for each residue, we use the reciprocal of GP-generated variance for the high-confidence predictions as weights to aggregate the phenotype values by using the following equation:

$$\hat{z} = \frac{\sum_i \frac{z_i}{\sigma_i^2}}{\sum_i \frac{1}{\sigma_i^2}} \qquad (7)$$

where $\hat{z}$ is the weighted mean value for each residue, $z_i$ is the predicted phenotype value at z axis for every value on the y axis, $\sigma_i^2$ is the GP-generated variance for each prediction. We repeat this process for all the residues. The IVW averaged mean values for all the residues then are mapped as barcodes or mapped on AAT structures (PDB:3NE4[67] for AAT monomer, PDB:2D26[76] for the AAT-elastase complex and PDB:3T1P[64] for polymer). All the atomic resolution structure presentations were produced with the software of PyMOL.

As Ordinary Kriging used in the above process does not quantify the uncertainty in the model parameters, we also used Bayesian Kriging to take account for the error introduced by the estimation of the variogram model. Specifically, we used the bayesGeostatExtact() function in the spBayes package[59–62]. We found that the phenotype landscape predictions generated by Ordinary Kriging were highly correlated with those generated by Bayesian Kriging (Supplementary Fig. 11a, Pearson's $r = 0.99$). The uncertainty values assessed by

Bayesian Kriging are generally larger than those generated by Ordinary Kriging (Supplementary Fig. 11b), but they are also highly correlated (Supplementary Fig. 11b, Pearson's $r = 0.98$). The resulting landscapes between Ordinary Kriging and Bayesian Kriging are almost identical (Supplementary Fig. 11c, d), with minor differences observed for the uncertainty (Supplementary Fig. 11c, d, contour lines, Supplementary Fig. 11e). After the IVW, the weighted average function value for each residue is again highly correlated (Supplementary Fig. 11f, Pearson's $r = 0.98$), which generates almost identical functional structures (Supplementary Fig. 11g). As Bayesian Kriging is much more computationally expensive than Ordinary Kriging, we used Ordinary Kriging in this study, but these results suggest that more advanced techniques can be applied to VSP to understand the sequence-to-function-to-structure relationships.

## Statistical analyses

Experimental data showed as the mean ± SD from three or more than three independent experiments. The differences between two independent groups were analyzed using Student's $t$ test. A $P$ value < 0.05 was considered statistically significant. The $P$ value of the Pearson's $r$ value is performed using ANOVA test. The null hypothesis for the $P$ value of the Pearson correlation is that the correlation coefficient is not significantly different from 0. The $P$ value represents the probability that the correlation occurred by chance. We used $P < 0.05$ to reject the null hypothesis and conclude there is a significant Pearson correlation. To compare the Pearson's correlation coefficients, we used Fisher Z-transformation to convert the Pearson's correlation coefficients into approximate normally distributed variables. We then conducted a hypothesis test with a null hypothesis where the transformed coefficients are equal. We used $P < 0.05$ to reject the null hypothesis.

## Reporting summary

Further information on research design is available in the Nature Portfolio Reporting Summary linked to this article.

## Data availability

Source datasets for each figure are provided in the Source Data file. Due to large data size, source data for Figs. 3c, d and 4a and Supplementary Figs. 7c, d, 8e, f, and 11 are included in the Mendeley Data (https://doi.org/10.17632/nt59rwz6r6.1). PDB files used in this study include 3NE4, 2D26 and 3T1P. Source data are provided with this paper.

## Code availability

The input data, R-code scripts, and output files for the GP-based analysis have been deposited in the public Mendeley database with https://doi.org/10.17632/nt59rwz6r6.1. The files are under CC BY 4.0 license.

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

## Acknowledgements

Grant support provided by NIH HL095524, DK051870, AG070209, AG049665, HL141810, and HG10881 to W.E.B. We thank Dr. Gabriela Chiosis for initially providing PU-WS13 compound. P.Z. was supported by in part the Ara Parseghian Medical Research Foundation Fellowship. C.W. was in part supported by the Alpha-1 Foundation Fellowship. We thank Dr. Mark Brantly for providing the Huh7.5 AAT–/– knockout cell line. We thank Ms. Fang Xiao for assistance on the conservation scores for each residue of AAT and Mr. Liangxiao Li for the statistical comparison between correlations.

## Author contributions

P.Z., C.W., S.S., X.W., and W.E.B. contributed to the experimental design. P.Z., C.W., and S.S. performed the experiments; C.W. and P.Z. performed the computational analysis; C.W., P.Z., S.S., and W.E.B. generated the manuscript.

## Competing interests

The authors declare no competing interests.
