## [Peer Review File · Nature Communications]

Tracing Genetic Diversity Captures the Molecular Basis of Misfolding DiseaseReviewer #2 (Remarks to the Author):

The manuscript submitted by Zhao and colleagues presents an important study that tested the capability of a machine learning method to predict the impact of a single amino acids substitution on the misfolding of AAT and its response to Hsp90 management.

The machine learning method, which is based on Gaussian process to profile the spatial covariance relationship. The manuscript is well written and address a key problem in the field.

I believe the manuscript could be considered for publication on Nature Communication after addressing the minor comments reported below.

1) The author claims that their method generates evolutionary based spatial covariance (SCV) maps. What is exactly in this context the meaning of the term "evolutionary"? Is this meaning that the predicted damaging variants tend to correspond to conserved sites of the protein. If this is the case, can you verify if there is some correlation between predicted values and the conservation score?

2) From the description of the methodology, it is not clear if the method is sensitive to the different possible type of mutant amino acids. Can you provide estimation of the level of variability in the predictions for the 19 possible mutations for each position along the sequence?

3) The dataset used for training of machine learning methods to predict the impact of variants can be biased toward specific regions of the protein. For this reason, the application of LOCV can lead to an overestimation of the performance. I suggest verifying what is the level of performance leave-one-out is performed, keeping all the variants in the same protein region.

Reviewer #3 (Remarks to the Author):

The manuscript describes a study that combines several powerful and diverse techniques to explore how proteostasis systems not only monitor but can actually dictate, the fate of a newly synthesized secretory protein. The secretory alpha1-antitrypsin protein was the subject of the study because of the many known and studied inherited mutations that exist in the polypeptide, some of which can prevent secretion, hinder the NE activity, and/or can actually promote the polymerization of the molecule. One of the most intriguing observations is that the selective drug-induced inhibition of the ATPase activity of molecular chaperone grp94 is able to prevent the intracellular degradation of misfolded AATZ and allow its proper folding. This is intriguing because grp94 is known to be necessary to promote the intracellular degradation (quality control) of mutant AAT. It is also impressive that many controls were included to question, and then substantiate a tentative conclusion generated earlier in the study. Also, the presentation of data is clear and the figures are attractive!

Without going into detail about all the observations, the multi-faceted analysis led to the emergence of many significant observations, allowing the authors to extend their observations to propose a theory of how the different components of the genome and proteome co-evolved to regulate and monitor structure, function, and whether to deploy the gene product! The findings are of great significance to those who study the lung and liver diseases associated with plasma AAT deficiency. However, it should be noted that AAT deficiency has arguably functioned as the prototype for the study of numerous loss of function and gain of toxic function diseases. Also, I should mention that the combination of techniques were rigorously tested and established in a prior study published in Cell Chemical Biology.

In terms of the protein and diseases under question, I should mention that the generated data have begun to clear up some inconsistencies as to what causes the misfolded AAT to polymerize and the actual structure of the toxic polymer. Also, I am confident that the findings will aid in the investigation of a broad range of protein-conformational diseases such as neurological diseases, cystic fibrosis, Type 2 diabetes, cancer, and a host of other diseases. Therefore, I fully expect that

the use of these different technologies, the interpretation of results, and the rigorous questioning of the observations will serve as a great roadmap for others to further study the related fields mentioned above.

With all of this said, the only "potential" flaws in this study are twofold. FIRST: According to published genetic data generated from patients (Ref 81), the role of Man1b1 for degrading ATZ is shown to be most prominent, and possibly even limited to, infants rather than to the entire patient ZZ population. Therefore, the protein degradation system responsible for eliminating misfolded AAT might differ between neonates and adults. This should be noted in the manuscript.

SECOND: I suspect that the conclusions made by the authors are correct. However, the authors should at least consider and mention that at least some of their observations might have resulted as a cellular response to the induction of alternative proteostasis machinery rather than from the loss of HSP94 activity. Please refer to the article (Eletto et al. *Journal of Cell Science* 125, 4865–4875, 2012) which should be cited. In other words, the results in the present study (i.e., in terms of the diminished AAT polymerization, enhanced secretion, and/or NE inhibition) could result from the enhancement of alternative systems/chaperones, possibly similar (but not identical to) what the senior author recently published in *Cell Chemical Biology* where the induction of two molecular chaperones played a significant role. However, if this alternative idea is unlikely, at least in the authors' opinion, then this should be stated in the Discussion Section. With that said, I do not believe that the addition of this information will detract from their results.

Reviewer #4 (Remarks to the Author):

Zhao, Wong, Balch and colleagues present analysis of AAT mutations on function with and without inhibition of the ER chaperone Grp94. AAT is a protease inhibitor in humans that is excreted from liver cells and blocks the activity of neutrophil elastase. This function of AAT is important for maintaining the extracellular matrix and function of the lungs. Human individuals that are homozygous for certain AAT mutations are at elevated risk for chronic obstructive pulmonary disease from smoking or pollution. The authors identified 76 AAT variants that have been observed in the human population. 44 of these AAT variants have been associated with disease while the remainder are of unknown significance or considered benign. The authors introduced each variant into Huh7.5 cells (a liver derived cell line) that had endogenous AAT removed by CRISPR. They also performed analyses in IB3 cells that are lung epithelial derived and do not generate detectable AAT protein. They also confirmed a few individual mutant effects in patient derived hepatic cells. They assessed the function of AAT variants at multiple levels including: secreted AAT protein, intracellular AAT protein, and the ability of secreted AAT to inhibit NE. The authors distinguished monomeric from aggregated AAT in these assays using antibodies that preferentially bind to heat-induced aggregated AAT or purified monomeric AAT. Most variants decreased secreted levels and inhibition of NE. The authors tested an inhibitor of Grp94 on AAT function – it had little impact on WT NE inhibition, but increased the activity of roughly half of the AAT mutants tested. The authors used a machine learning approach in an attempt to understand the structural/biophysical features that determine AAT folding and function from their measurement on 76 variants. From the structural analyses, the authors conclude that formation of B-strand B of AAT is assisted by Grp94.

The Balch lab has a strong track record of cutting edge research on clinically relevant protein homeostasis, and the experimental work reported here appears excellent and reveals some exciting and interesting new findings related to AAT folding and the involvement of Grp94 in this process. The methods section and the Figures are well written and presented clearly. However, the machine learning results as well as much of the introduction, results, and discussion sections appear to be written with more of a high-profile bias (attempt to make the findings appear general and/or outstanding) rather than with clarity and objectivity. Many of the statements in these sections (some, but not all of which are listed below) appear to be without strong support. To me,

the findings of the experimental work are outstanding and merit publication on their own merits. The observation that Grp94 inhibition increases the secretion and function of many AAT variants is new and important. That these variants are generally not rescued by Grp94 knockdown is fascinating and while not fully understood, ought to be discussed in greater length (see comments below). I have listed my major concerns below, all of which I believe can be readily addressed. With modifications, this work would merit publication in Nature Comm from my perspective.

Major concerns.

1. The authors should clearly describe the expression of AAT in their experiments and how they compare to endogenous expression. They describe using pCDNA3.1 in the methods section which utilizes a viral promoter. In the results section, the authors should describe what level of AAT protein this yields compared to endogenous expression of AAT in liver cells? Does expression from the system used in this paper lead to different levels of aggregated protein compared to endogenous expression? How does this impact interpretation of results with mutants? Presumably the authors have looked at this in prior work, but it should be summarized here as it is important for assessing the results.

2. I found it fascinating that Grp94 inhibition leads to increased AAT function but that Grp94 knockdown does not. It would be tremendously helpful to include a discussion of the impacts of Grp94 or Hsp90 inhibition and knockdown on other clients. Is this a typical or unusual pattern and what have other studies concluded regarding the mechanism of Hsp90 inhibition or knockdown on substrates. This is relevant to the authors speculation of the role of alterations in Grp94 ATP-driven conformations. In my view this section of the discussion should be expanded and the authors meaning clarified. The authors should clearly describe why they think inhibition is different than knockdown and whether this is common or unusual among clients.

3. The authors refer to design with regard to AAT properties, but even after multiple readings, I was not clear what they mean by design. To me, and many others, design implies intention. Referring to the outcomes of an evolutionary process as design, seems confusing and unfortunately prone to propaganda by people with faith-based agendas. I urge the authors to consider omitting design with regards to evolution or products of evolution. As far as I can tell, design can be left out and the text improved by shortening in many places ... as an example (one of many): Line 199 ... "These results suggest that Grp94 impacts the AAT fold design features that contribute to both its monomer secretion and NE inhibitory activity, including the prominent AAT-Z and AAT-S alleles contributing to clinical disease." Could be re-written as ... These results indicate that Grp94 impacts the folding and secretion of active AAT, including the AAT-Z and AAT-S alleles that contribute to disease.

4. Even after multiple readings, I could not appreciate the value of the machine learning approaches. As this is targeted for a general audience (as opposed to a journal more focused on computational approaches), I encourage the authors to explain their key findings with greater clarity (and perhaps reduce the length of discussion and the number of figures on GP. In particular, it was not clear to me if the GP approach that they utilized took into account the type of mutation (not just the position). If it doesn't take into account the type of mutation, then is that saying that all mutations at the same position will have similar effects? If this is the case, I would want to see a far more in depth discussion of these implications. Along the same lines, it was not clear if GP takes into account knowledge of the AAT protein structure? The authors should make clear why GP was chosen including what were there goals and what alternative approaches could have been taken, and why GP was chosen. In my view, the machine learning description is insufficient/inappropriate for a general audience journal in its current state.

5. How does separating mutations into location and conservation (e.g. BLOSUM) predict functional outcomes. For example are the mutations with the largest disruption the most dramatic amino acid changes at buried positions in the structure?

6. As I understand the machine learning approach, the authors used it to make predictions about the impacts of AAT mutations that have not yet been subjected to experimental evaluation. In this

case, it would dramatically improve the manuscript if the authors experimentally assessed some of these mutations as a way to evaluate the predictions. Tested predictions are far more valuable than untested ones.

7. How were class cutoffs decided (see results lines 170-181)? Defining 75% activity of WT as loss of function seems severe. Ideally these cutoffs should be linked to phenotypes, or at least described how they might relate to phenotype. Is 75% of WT activity likely to cause disease?

8. Line 344-349. What does this sentence mean? Does it simply mean that these are locations where mutations can disturb the folding in a way that inhibition of Grp94 can partially rescue? Are the authors suggesting that these are structurally special sites? If so, in what way are they special? Are the nucleation sites for critical early or rate limiting steps in folding? Might they relate to nucleation sites for folding akin to those identified by the Fersht lab using Psi analyses?

9. Line 347 is challenging to read: "how evolution evolves" perhaps could be simplified to, "how noun evolves"

10. Line 422-423 – Does this mean that the authors believe intracellular polymer can be secreted directly? Why is this more likely than secreted monomer aggregating? What is the reasoning here?

11. Line 424-427. Since the authors have measures of intracellular aggregation, it is puzzling that they use measures of extracellular aggregation in order to investigate potential impacts on liver disease (that is believed to be caused by intracellular aggregates). Even if intracellular and extracellular aggregates correlate, wouldn't it be better to use intracellular measurements. They do not correlate perfectly and wouldn't the intracellular measurements be more relevant?

12. What is the level of inhibition of Grp94 in these studies? In other words, what is the concentration of inhibitor relative to K_i ? The authors mention in a few places that they are slowing ATPase activity – but not by how much? Is it slowed by half, 10-fold, 100-fold?? This has important implications for their conclusions regarding slowed ATPase activity of Grp94.

13. Line 444-448. Are the authors proposing that Grp94 inhibited with PU-WS13 does this? How? This seems highly speculative as the authors do not have direct structural observations – Does Grp94 inhibited with PU-WS13 bind to AAT? Or disease variants?

14. Lines 497-503. This seems overstated.

15. What do the authors think is going on with variants in Fig 1D with elevated ratios of NE inhibition to monomer secretion. Do these variants have greater specific activity (suggesting that the WT is either maladaptive or under balancing selection)? Or is some other mechanism at work?

16. In Fig 1G, the relationship seems bi-phasic – it would be interesting to know why this might be the case. Is this approaching full inhibition levels, and a simple outcome of the experimental setup?

17. Line 527: Targeting this cooperative folding provides a new mechanism for drug development. This seems conceptually similar to the idea of chemical chaperones from Jeff Kelly. The authors should clarify what they mean by a new mechanism for drug development – what is this mechanism and how does it compare to other related concepts such as chemical chaperones being developed for other protein folding diseases such as Gauchier's and Cystic Fibrosis.

Minor concerns/thoughts:

1. Perhaps it is beyond the scope of this work, but it seems like exploring the role of ubiquitination/proteasome degradation would be worthwhile in future directions.

2. Line 228 – what is the statistical test used to compare correlations? Both correlations are

statistically significant, but here authors are talking about changes in R being significant – what test was used for this?

REVIEWER COMMENTS

Reviewer #2 (Remarks to the Author):

The manuscript submitted by Zhao and colleagues presents an important study that tested the capability of a machine learning method to predict the impact of a single amino acids substitution on the misfolding of AAT and its response to Hsp90 management. The machine learning method, which is based on Gaussian process to profile the spatial covariance relationship. The manuscript is well written and address a key problem in the field. I believe the manuscript could be considered for publication on Nature Communication after addressing the minor comments reported below.

1) The author claims that their method generates evolutionary based spatial covariance (SCV) maps. What is exactly in this context the meaning of the term "evolutionary"?

The reason we use 'evolution based spatial covariance' is that we use the collective of genetic variants in the world-wide population and their associated cellular phenotypes to generate the SCV maps. Each variant residue reports on the evolutionary basis of the protein fold across the human population showing how the changes in the polypeptide sequence affects its function in the host. Variation can also report on the mechanism(s) of how proteostasis manages the residue-by-residue conversion of the form and function of protein fold to generate the healthy state and how that state deviates in disease in response to variation. We have modified the text to provide the rationale (lines 50-52 highlighted in blue and underlined).

Is this meaning that the predicted damaging variants tend to correspond to conserved sites of the protein. If this is the case, can you verify if there is some correlation between predicted values and the conservation score?

Conservation scores derived from the alignment of ancestral sequences have no direct information regarding various cellular phenotypes. Furthermore, conserved residues across ancestral sequences are usually also conserved across the human population, so they do not report how the human population regulates the various functional features of a protein fold to drive the fitness of individuals. We used 109 different AAT sequences from multiple species and generated the conservation scores for each residue using Shannon entropy scores. The conservation score has a weak negative correlation with the SCV-derived NE inhibitory activity score (Pearson's $r = -0.16$, $p = 0.001$), suggesting that the conserved residues have a weak trend to confer defective NE inhibitory activity levels. Moreover, the conservation score has no significant correlation with either monomer secretion (Pearson's $r = -0.1$, $p = 0.05$) or intracellular polymer (Pearson's $r = 0.03$, $p=0.5$), indicating that the conservation score does not provide information for detailed cellular phenotypes. The differences between conservation scores and our approach are now presented in Supplementary **Fig. S9** and added to the **Discussion** at lines 543 to line 547 (underlined).

2) From the description of the methodology, it is not clear if the method is sensitive to the different possible type of mutant amino acids. Can you provide estimation of the level of variability in the predictions for the 19 possible mutations for each position along the

sequence?

The purpose of the GP method is not to predict every possible mutation for each residue in the sequence, but rather to use the sparse collection of variants in the natural population reporting on function, particularly the variants found in the patients driving disease, to compute the probability of every amino acid residue in the polypeptide chain to generate a given cellular phenotype as described in our pioneering 2018 (Cell Reports, reference 1) and 2022 papers (Cell Structure, reference 21)- as well as 5 others that describe how epigenetics and other proteostasis modulators can manipulate SCV landscapes for correction of cystic fibrosis, Niemann-Pick C1 as well as AAT. In the current manuscript, we use variants as tools to understand the mechanisms of proteostasis management, in this case, the rescue effect of GRP94 regulator for AAT misfolding and dysfunction on a residue-by-residue basis using GP. In our dataset, we do have different types of amino acid mutants found at the same residue location, for example, G344R/G344E, A360P/A360T, P393L/P393S/P393T. They have different spatial positions in the SCV landscapes according to their different cellular phenotype values used as input. We have expanded our description of the use inverse variance weighting (IVW), as more completely described in lines 333 to 355 (underlined) (and originally developed and described in Cell Structure, 2022, reference 21), to average the phenotype values for different variants for each residue position to compute the most likely role of that residue contributing to various cellular phenotypes seen in the AATD population as well as captured by biochemical metrics. In so doing, GP quantitatively helps guides us in our global understanding of the general principles governing the AAT polypeptide architecture.

3) The dataset used for training of machine learning methods to predict the impact of variants can be biased toward specific regions of the protein. For this reason, the application of LOCV can lead to an overestimation of the performance. I suggest verifying what is the level of performance leave-one-out is performed, keeping all the variants in the same protein region.

Because we use genetic variants found distributed across the world-wide population, particularly variants reported to cause disease in the patients, they do have different distributions along different regions of the protein likely reflecting hotspots in fold architecture, and therefore help us to define the diverse roles of different regions of the fold in response to evolutionary changes in sequence. Depending on the different spatial distribution patterns of the input variants, Gaussian process-based machine learning can report different confidence levels for each prediction. Protein regions that harbor more variants in the population usually have a higher associated prediction confidence than regions with fewer variants. The prediction confidence is displayed as contour lines in the landscapes as described from lines 280 to line 289 (underlined). We normally validate the value of the landscape predictions for the entire sequence using leave-one-out cross-validation. More specifically, and to address Reviewer's concern, we computed the leave-one-out cross-validation for the key select regions 1-139 (Pearson's $r=0.78$, $p=1.7 \times 10^{-5}$), 140-278 (Pearson's $r=0.7$, $p=0.003$) and 279-418 (Pearson's $r=0.73$, $p=9 \times 10^{-7}$) separately for the NE inhibitory activity landscape (**Fig. 3C**). The slightly lower prediction accuracy in residues 140-278 is consistent with lower prediction confidence in this region as illustrated by lower high-confidence contour lines in the phenotype landscape (**Fig. 3C**). Nevertheless, each region shows strong correlations between the predictions and

experimental values, illustrating the accuracy of the modeling across the entire polypeptide chain is high. We added the description of these results in lines 292 to line 296 (underlined).

Reviewer #3 (Remarks to the Author):

The manuscript describes a study that combines several powerful and diverse techniques to explore how proteostasis systems not only monitor but can actually dictate, the fate of a newly synthesized secretory protein. The secretory alpha1-antitrypsin protein was the subject of the study because of the many known and studied inherited mutations that exist in the polypeptide, some of which can prevent secretion, hinder the NE activity, and/or can actually promote the polymerization of the molecule. One of the most intriguing observations is that the selective drug-induced inhibition of the ATPase activity of molecular chaperone grp94 is able to prevent the intracellular degradation of misfolded AATZ and allow its proper folding. This is intriguing because grp94 is known to be necessary to promote the intracellular degradation (quality control) of mutant AAT. It is also impressive that many controls were included to question, and then substantiate a tentative conclusion generated earlier in the study. Also, the presentation of data is clear and the figures are attractive!

Without going into detail about all the observations, the multi-faceted analysis led to the emergence of many significant observations, allowing the authors to extend their observations to propose a theory of how the different components of the genome and proteome co-evolved to regulate and monitor structure, function, and whether to deploy the gene product! The findings are of great significance to those who study the lung and liver diseases associated with plasma AAT deficiency. However, it should be noted that AAT deficiency has arguably functioned as the prototype for the study of numerous loss of function and gain of toxic function diseases. Also, I should mention that the combination of techniques were rigorously tested and established in a prior study published in Cell Chemical Biology.

In terms of the protein and diseases under question, I should mention that the generated data have begun to clear up some inconsistencies as to what causes the misfolded AAT to polymerize and the actual structure of the toxic polymer. Also, I am confident that the findings will aid in the investigation of a broad range of protein-conformational diseases such as neurological diseases, cystic fibrosis, Type 2 diabetes, cancer, and a host of other diseases. Therefore, I fully expect that the use of these different technologies, the interpretation of results, and the rigorous questioning of the observations will serve as a great roadmap for others to further study the related fields mentioned above.

1) With all of this said, the only "potential" flaws in this study are twofold. FIRST: According to published genetic data generated from patients (Ref 81), the role of Man1b1 for degrading ATZ is shown to be most prominent, and possibly even limited to, infants rather than to the entire patient ZZ population. Therefore, the protein degradation system responsible for eliminating misfolded AAT might differ between neonates and adults. This should be noted in the manuscript.

We thank the reviewer for this insight. We have now included this excellent point in the **Discussion** at lines 568-572 (underlined). It illustrates very nicely how the proteostasis can sense aberrant SCV relationships that change during development and aging.

2) I suspect that the conclusions made by the authors are correct. However, the authors should at least consider and mention that at least some of their observations might have resulted as a cellular response to the induction of alternative proteostasis machinery rather than from the loss of HSP94 activity. Please refer to the article (Eletto et al. *Journal of Cell Science* 125, 4865–4875, 2012) which should be cited. In other words, the results in the present study (i.e., in terms of the diminished AAT polymerization, enhanced secretion, and/or NE inhibition) could result from the enhancement of alternative systems/chaperones, possibly similar (but not identical to) what the senior author recently published in *Cell Chemical Biology* where the induction of two molecular chaperones played a significant role. However, if this alternative idea is unlikely, at least in the authors' opinion, then this should be stated in the Discussion Section. With that said, I do not believe that the addition of this information will detract from their results.

Consistent with the reference suggested by the reviewer, we observed an increase of BiP expression in response to silencing of GRP94 (new **Fig. S10F**). However, unlike a general Hsp90 inhibitor referenced above that can inhibit **both** GRP94 and Hsp90, we did not observe a significant increase of BiP when we added the specific GRP94 inhibitor, PU-WS13 at 1 μ M (new **Fig. S10F**). This result is consistent a previous article (reference 94) that shows PU-WS13 treatment does not increase BiP levels in macrophages. Therefore, we think the correction of AAT-Z secretion is more likely directly linked to the PU-WS13 bound form of GRP94. However, it does not rule out the possibility that other proteostasis components may be involved in the rescue of AAT-Z and have included the suggested reference (line 592). For example, we found that there is an increase of binding between ERGIC-53 and AAT-Z by PU-WS13 treatment, which may account for improved secretion of AAT-Z (new **Fig. S10G**). We have added these points at line 581-595 (underlined) in the **Discussion**.

Reviewer #4 (Remarks to the Author):

Zhao, Wong, Balch and colleagues present analysis of AAT mutations on function with and without inhibition of the ER chaperone Grp94. AAT is a protease inhibitor in humans that is excreted from liver cells and blocks the activity of neutrophil elastase. This function of AAT is important for maintaining the extracellular matrix and function of the lungs. Human individuals that are homozygous for certain AAT mutations are at elevated risk for chronic obstructive pulmonary disease from smoking or pollution. The authors identified 76 AAT variants that have been observed in the human population. 44 of these AAT variants have been associated with disease while the remainder are of unknown significance or considered benign. The authors introduced each variant into Huh7.5 cells (a liver derived cell line) that had endogenous AAT removed by CRISPR. They also performed analyses in IB3 cells that are lung epithelial derived and do not generate detectable AAT protein. They also confirmed a few individual mutant effects in patient derived hepatic cells. They assessed the function of AAT variants at multiple levels including: secreted AAT protein, intracellular AAT protein, and the ability of secreted

AAT to inhibit NE. The authors distinguished monomeric from aggregated AAT in these assays using antibodies that preferentially bind to heat-induced aggregated AAT or purified monomeric AAT. Most variants decreased secreted levels and inhibition of NE. The authors tested an inhibitor of Grp94 on AAT function – it had little impact on WT NE inhibition, but increased the activity of roughly half of the AAT mutants tested. The authors used a machine learning approach in an attempt to understand the structural/biophysical features that determine AAT folding and function from their measurement on 76 variants. From the structural analyses, the authors conclude that formation of B-strand B of AAT is assisted by Grp94.

The Balch lab has a strong track record of cutting edge research on clinically relevant protein homeostasis, and the experimental work reported here appears excellent and reveals some exciting and interesting new findings related to AAT folding and the involvement of Grp94 in this process. The methods section and the Figures are well written and presented clearly. However, the machine learning results as well as much of the introduction, results, and discussion sections appear to be written with more of a high-profile bias (attempt to make the findings appear general and/or outstanding) rather than with clarity and objectivity. Many of the statements in these sections (some, but not all of which are listed below) appear to be without strong support. To me, the findings of the experimental work are outstanding and merit publication on their own merits. The observation that Grp94 inhibition increases the secretion and function of many AAT variants is new and important. That these variants are generally not rescued by Grp94 knockdown is fascinating and while not fully understood, ought to be discussed in greater length (see comments below). I have listed my major concerns below, all of which I believe can be readily addressed. With modifications, this work would merit publication in Nature Comm from my perspective.

Major concerns.

1. The authors should clearly describe the expression of AAT in their experiments and how they compare to endogenous expression. They describe using pCDNA3.1 in the methods section which utilizes a viral promoter. In the results section, the authors should describe what level of AAT protein this yields compared to endogenous expression of AAT in liver cells? Does expression from the system used in this paper lead to different levels of aggregated protein compared to endogenous expression? How does this impact interpretation of results with mutants? Presumably the authors have looked at this in prior work, but it should be summarized here as it is important for assessing the results.

In our experimental system, the transient transfection of AAT-Z generates an ~1.2-fold of monomer secretion and 1.3-fold of polymer secretion when compared with AAT-ZZ with endogenous promoter in the hepatic lineages prepared from iPSCs (iHepZZ) derived from a homozygous AAT-ZZ patient (new Fig. S4A). Consistent with the results from the transient overexpression of AAT-Z in Huh7^{null} and IB3^{null} cells, PU-WS13 increased the secreted monomer and NE inhibitory activity, while reducing the intracellular polymer in

the iHepZZ cells (**Fig. 2D**; **Fig. S4B-F**). These results suggest that PU-WS13 corrects AAT-ZZ in the context of its endogenous promoter. We added a comment for this point in lines 217-224 (underlined) and a new Supplemental **Fig. S4A**.

2. I found it fascinating that Grp94 inhibition leads to increased AAT function but that Grp94 knockdown does not. It would be tremendously helpful to include a discussion of the impacts of Grp94 or Hsp90 inhibition and knockdown on other clients. Is this a typical or unusual pattern and what have other studies concluded regarding the mechanism of Hsp90 inhibition or knockdown on substrates. This is relevant to the authors speculation of the role of alterations in Grp94 ATP-driven conformations. In my view this section of the discussion should be expanded and the authors meaning clarified. The authors should clearly describe why they think inhibition is different than knockdown and whether this is common or unusual among clients.

This is an important point and reveals the differential impact that co-chaperones can have on the protein fold. For example, inhibition of GRP94 through PU-WS13 has been shown to reduce the stability of HER2 in a tumor-specific manner. Here, siGRP94 mimics the effect of PU-WS13 to destabilize HER2. However, in our study as highlighted above, we found that whereas PU-WS13 rescues the folding, secretion, and function of AAT-Z, silencing through siGRP94 did not rescue AAT-Z defects. What we are seeing is the dynamic features of GRP94 function affecting its potential differential roles in promoting stability (AAT-Z) vs degradation (HER2), suggesting that the correction effect we observe is a part of a larger programmable system in which the GRP94 ATPase activity can be reprogrammed by co-chaperones to achieve optimal function in response to variation in the context of the local environment. In other words, the ATP-dependent chaperone cycle of GRP94 is tuned to the individual dynamics of the protein fold defined by GP based SCV relationships. We now more thoroughly developed this important point in lines 581-594 (underlined) as suggested.

3. The authors refer to design with regard to AAT properties, but even after multiple readings, I was not clear what they mean by design. To me, and many others, design implies intention. Referring to the outcomes of an evolutionary process as design, seems confusing and unfortunately prone to propaganda by people with faith-based agendas. I urge the authors to consider omitting design with regards to evolution or products of evolution. As far as I can tell, design can be left out and the text improved by shortening in many places ... as an example (one of many): Line 199 ... “These results suggest that Grp94 impacts the AAT fold design features that contribute to both its monomer secretion and NE inhibitory activity, including the prominent AAT-Z and AAT-S alleles contributing to clinical disease.” Could be re-written as ... These results indicate that Grp94 impacts the folding and secretion of active AAT, including the AAT-Z and AAT-S alleles that contribute to disease.

Good point. We have removed the word ‘design’ in the text in accord with the reviewer’s concerns.

4. Even after multiple readings, I could not appreciate the value of the machine learning approaches. As this is targeted for a general audience (as opposed to a journal more focused on computational approaches), I encourage the authors to explain their key findings with greater clarity (and perhaps reduce the length of discussion and the number of figures on GP. In particular, it was not clear to me if the GP approach that they utilized took into account the type of mutation (not just the position). If it doesn't take into account the type of mutation, then is that saying that all mutations at the same position will have similar effects? If this is the case, I would want to see a far more in depth discussion of these implications. Along the same lines, it was not clear if GP takes into account knowledge of the AAT protein structure? The authors should make clear why GP was chosen including what were there goals and what alternative approaches could have been taken, and why GP was chosen. In my view, the machine learning description is insufficient/inappropriate for a general audience journal in its current state.

As stated in the response to Reviewer 2, we do have different types of amino acid mutants on the same residue location in our dataset, for example, G344R/G344E, A360P/A360T, P393L/P393S/P393T. They have different spatial positions in the SCV landscapes according to their different cellular phenotype values. We use inverse variance weighting (IVW), as described in the manuscript (lines 333-355, underlined), to average the phenotype values for different variants on each residue position to compute the most likely role of that residue contributing to various cellular phenotypes. Nature works at a probabilistic level with defined uncertainty so we take advantage of that when considering what is the most likely outcome. The reason we chose GP, as thoroughly detailed in the 2018 pioneering SCV paper (Cell Reports, reference 1) and in our 2022 Triangulation paper (Cell Structure, reference 21) where we precisely correlated the GP analysis with structural distances at atomic resolution, as well as a recently published paper in a Cell Patterns (reference 123) not yet listed in PubMed (doi.org/10.1016/j.patter.2023.100800)) where we define the GP predictive ability on a allele-by-allele basis to understand the evolution of the SARS-CoV-2 pandemic through SCV principle thinking, is that GP can rigorously model uncertainty to allow us to define the 'most likely' contribution of any residue in the AAT sequence in function in health and disease. We do not use AAT protein structure as input data in GP modeling- but rather we use the structural data to validate the short- and long-range correlations we observed through GP prediction, for example, the N-terminal to C-terminal cooperative folding of AAT in response to the GRP94 inhibitor. Structures only provide a single snapshot- whereas GP captures the ground truth spatial-temporal dynamics relating sequence-to-function-to-structure in a living cell and how each residue can impact its biology. We have expanded the description of the approach as suggested to explain the power of the GP approach more thoroughly in capturing the functional architecture of the AAT fold (lines 333-355 (underlined)).

5. How does separating mutations into location and conservation (e.g., BLOSUM) predict functional outcomes. For example are the mutations with the largest disruption the most dramatic amino acid changes at buried positions in the structure?

As stated in the response to Reviewer 2, the conservation score cannot provide information on various cellular phenotypes of AAT variants (new **Fig. S9**). Some buried residues in the structure, for example, residues in s5A in β -sheet A are important for AAT function (**Fig. 4D**). However, the most responsive residues to GRP94 management are the residues that are shuttled around in the polymerization process, for example, regions around s5B and s6B in β -sheet B, as well as regions around the beginning of the reaction loop. Variants in the human population, particularly the patient population, can inform on the dynamics of ‘static’ structural snapshots that are important in responding to proteostasis and subjected to sequence optimization by the human population- for better (improved health) or worse (disease)- through natural selection.

6. As I understand the machine learning approach, the authors used it to make predictions about the impacts of AAT mutations that have not yet been subjected to experimental evaluation. In this case, it would dramatically improve the manuscript if the authors experimentally assessed some of these mutations as a way to evaluate the predictions. Tested predictions are far more valuable than untested ones.

The major purpose of the machine learning approach is to use the variation in the world-wide population, particularly the variants found in the patient population, to compute the most probable functional feature of each amino acid residue as mentioned above. The methods generate high confidence predictions (generally 95% of the protein sequence) of the most likely function with assigned uncertainty for all AAT residues and, in particular, mutations that were not included in the training dataset. We have tested the model's accuracy by using leave-one-out cross-validation throughout this and every published manuscript to date. For example, using the same machine learning approach, we successfully capture the response of an uncharacterized mutant to ATF6 activator (**Fig. S7** in the reference 34). These results suggest that the GP approach can accurately predict a variant for which the computational model was not trained, consistent with the strong correlation between predicted values and actual measurements using leave-one-out cross-validation. While we fully agree that tested predictions can potentially be more valuable than untested ones, we take advantage of the fact that by incorporating function using entire sequence of SCV relationships dictated by each residue in the polypeptide chain (based on the rules of natural selection responsible for human evolution) we are getting closer to the ground truth than experiments performed in the artificial conditions of the laboratory. In other words, nature has already done the experiment for us and by learning from her rules we are more successful in assessing the role of each residue in response to variation, the environment and to therapeutics. GP regression used herein was derived from geostatistics (spatial analysis) where it is an exceptionally powerful tool to find value in complex landscapes (see pioneering Cell Report (2018) paper (reference 1) where you cannot ‘repeat’ the experiment in the laboratory. We are taking advantages of the many successes of the GP mathematical platform originally applied to geology (i.e., finding oil) but have now applied it to biology.

7. How were class cutoffs decided (see results lines 170-181)? Defining 75% activity of

WT as loss of function seems severe. Ideally these cutoffs should be linked to phenotypes, or at least described how they might relate to phenotype. Is 75% of WT activity likely to cause disease?

There are two reasons for choosing 75% as the class cutoff: 1) In the 40 AAT variants that have been reported to cause **clinical disease phenotypes**, 33 of them (>80%) have NE inhibitory activity lower than 75% of WT activity and 2) The mean of the predicted NE inhibitory of all residues of AAT is around 73%. Therefore, we set the activity cutoff as 75%. We have clarified this important point in the manuscript by adding these thoughts in lines 172 to 176 (underlined).

8. Line 344-349. What does this sentence mean? Does it simply mean that these are locations where mutations can disturb the folding in a way that inhibition of Grp94 can partially rescue? Are the authors suggesting that these are structurally special sites? If so, in what way are they special? Are the nucleation sites for critical early or rate limiting steps in folding? Might they relate to nucleation sites for folding akin to those identified by the Fersht lab using Psi analyses?

This is an excellent point. We have modified the description to address this concern as well as taken the opportunity to reference Fersht Phi angle concept (new reference 62) based on energetics to provide a potential rationale for the observed function of GRP94. The results suggest that perhaps ‘hotspots’ affecting both short and long-range interactions contribute to the functional response to GRP94 management. As indicated for polymer responses to PU-WS13, β -sheet B is important for the C-terminal model of the polymerization process. Thus, managing the long-range assembly of N1, M2, and C3 domains in β -sheet B through GRP94 inhibition is critical for rescuing AAT-Z folding, secretion, and function. We have adjusted the text in lines 359-364 (underlined) to address this point.

9. Line 347 is challenging to read: “how evolution evolves” perhaps could be simplified to, “how noun evolves”

We have removed the phrase ‘how evolution evolves’ in the sentence.

10. Line 422-423 – Does this mean that the authors believe intracellular polymer can be secreted directly? Why is this more likely than secreted monomer aggregating? What is the reasoning here?

The correlation between intracellular polymer load and extracellular secreted polymer in media and serum is well-established as documented in references 32,65-67. PU-WS13 modifies the activity of GRP94 in the ER to promote secretion and we note that the changes for the intracellular AAT-Z polymer are highly correlated to the secreted polymer. While

certainly over the long-term AAT-Z could polymerize in serum, there is no evidence to suggest this is the case. Our conclusions are based on an extensive literature on the topic.

11. Line 424-427. Since the authors have measures of intracellular aggregation, it is puzzling that they use measures of extracellular aggregation in order to investigate potential impacts on liver disease (that is believed to be caused by intracellular aggregates). Even if intracellular and extracellular aggregates correlate, wouldn't it be better to use intracellular measurements. They do not correlate perfectly and wouldn't the intracellular measurements be more relevant?

Secreted polymer can be directly measured from patients' plasma, while the intracellular polymer in the liver hepatocytes is very hard to measure (i.e., would require routine liver biopsy). So, the secreted polymer is a more useful biomarker to report liver disease phenotypes for clinical management (see references above) of disease, a metric we have incorporated into the current interpretation of our results and analyses that faithfully reproduce what is seen in the patient.

12. What is the level of inhibition of Grp94 in these studies? In other words, what is the concentration of inhibitor relative to K_i ? The authors mention in a few places that they are slowing ATPase activity – but not by how much? Is it slowed by half, 10-fold, 100-fold?? This has important implications for their conclusions regarding slowed ATPase activity of Grp94.

The IC_{50} of PU-WS13 for GRP94 inhibition is 0.22 μM , while the IC_{50} of PU-WS13 for inhibiting HSP90 α , HSP90 β , and TRAP1 are 27.3 μM , 41.8 μM , and 7.3 μM , respectively (reference 23 in the manuscript). We have tested PU-WS13 in different doses in Huh7.5 cells (**new Fig. S10A-E**). Our results showed that 1 μM PU-WS13 treatment achieved the maximum corrections for Z-AAT NE activity, monomer secretion, and reducing polymers. More PU-WS13 does not increase the correction effect anymore. Therefore, we used 1 μM PU-WS13 treatment for the experiments in the study. We didn't specifically measure the level of GRP94 activity in the cell, but according to its IC_{50} , 1 μM PU-WS13 treatment should block most of GRP94 activity while avoiding potential off-target effect on other Hsp90 family members. We have added this information to the main text of the manuscript to clarify the rationale for use of Grp94 in lines 190-195.

13. Line 444-448. Are the authors proposing that Grp94 inhibited with PU-WS13 does this? How? This seems highly speculative as the authors do not have direct structural observations –

We agree it is speculative based on the functional polymer structure (**Fig. 6G-I**) which is why we simply state it is 'consistent' with the C-terminal polymerization model.

Does Grp94 inhibited with PU-WS13 bind to AAT? Or disease variants? Based on our pull-down experiments, GRP94 treated by PU-WS13 still binds to AAT-Z,

but binding is reduced when compared with the level in untreated cells. Moreover, PU-WS13 improves the binding of AAT-Z to ERGIC53, consistent with the observation that PU-WS13 increases the secretion of AAT-Z. We now included this as figure (Supplemental **Fig. S10G**).

14. Lines 497-503. This seems overstated.

Agreed. We have removed the speculation about the utility of GP as a standard model in the **Results**, but have retained the comment in the last paragraph of the **Discussion** to consistent with its speculative nature, but reflecting our ability to use the GP approach across a broad range proteins (references 1, 17-21) and viral sequences (reference 123) (see lines 613-621 (underlined)).

15. What do the authors think is going on with variants in Fig 1D with elevated ratios of NE inhibition to monomer secretion. Do these variants have greater specific activity (suggesting that the WT is either maladaptive or under balancing selection)? Or is some other mechanism at work?

As pointed out, we do observe an increase in specific activity for select regions, as described in **Fig 7**, in response to GPR94 inhibition. We recently highlighted this effect when we examined the impact of ATF6 regulators on AAT folding, secretion and NE activity (see reference 34, **Fig. 7B**). Here, we documented a change in specific activity across a broad spectrum of variants is dramatic in response to a more global activation of the unfolded protein response (UPR) in which GRP94 is one of the components upregulated. There are also regions showing reduced specific activity (i.e., monomer secretion is improved more than the rescue of activity). These results suggest that sequence-to-function-to-structure relationships are actively managed by proteostasis. The exact mechanism at work remains speculation but relates to our increasing awareness of the role of proteostasis in mitigating ‘conversion’ as observed in prion disease. These results raise the possibility that the folding environment impacted by GRP94 in ER can have a major impact on evolutionary processes driving natural selection as has been suggested previously by Lindquist and others (references 105-121; lines 613-622 (underlined)) as highlighted in the **Discussion** that are now quantitatively captured by GP based SCV relationships dictating management of fold function by proteostasis (lines 613-622 (underlined)).

16. In Fig 1G, the relationship seems bi-phasic – it would be interesting to know why this might be the case. Is this approaching full inhibition levels, and a simple outcome of the experimental setup?

It is likely due to the different sensitivity of the assays. For example, the NE inhibitory assay is saturated once it reaches WT-level (to improve sensitivity for lower levels), while the monomer assay is not. The correlation was linear in the range of $0 < x < 1$ and $0 < y < 1$, suggesting that the assay is good for capturing the changes in variant function with phenotype levels between that of Z and WT.

17. Line 527: Targeting this cooperative folding provides a new mechanism for drug development. This seems conceptually similar to the idea of chemical chaperones from Jeff Kelly. The authors should clarify what they mean by a new mechanism for drug development – what is this mechanism and how does it compare to other related concepts such as chemical chaperones being developed for other protein folding diseases such as Gaucher’s and Cystic Fibrosis.

Whereas past studies have focused on chemical kinetic stabilization of the fold to prevent diseases of misfolding including transthyretin and immunoglobulin light chain amyloidosis, Gaucher’s and cystic fibrosis, SCV provides a fresh approach to understand more completely the complex features of cooperative folding and fold architecture to develop new insights into repairing challenging sequence-to-function-to-structure relationships that prevent misfolding while maintaining activity. As stated in the manuscript, simply stabilizing the structure to inhibit the polymerization does not work as it also blocks the function of AAT (reference 40). The SCV approach used for AAT (and cystic fibrosis) uses the relationships defined by genetic variants found in the population to differentiate the contribution of a residue to aggregation vs function at a residue-by-residue level for the entire polypeptide of AAT. SCV allows us to discover new target sites for reducing the polymer while increasing the function at the same time and is thus tuned to the functional dynamics of the fold that would otherwise be compromised by kinetic stabilizers. This is the next step in more pro-actively understanding what you need to fix before you try to fix it.

Minor concerns/thoughts:

1. Perhaps it is beyond the scope of this work, but it seems like exploring the role of ubiquitination/proteasome degradation would be worthwhile in future directions.

Absolutely. Given that degradation is not the focus of this paper, we have only briefly mentioned degradation/autophagy pathways (lines 576-580).

2. Line 228 – what is the statistical test used to compare correlations? Both correlations are statistically significant, but here authors are talking about changes in R being significant – what test was used for this?

We used Fisher Z-transformation to convert Pearson’s correlation coefficients into approximately normally distributed variables. Then we conducted a hypothesis test with a null hypothesis where the transformed coefficients are equal. This yields a p-value = 0.02, suggesting the correlation under the DMSO condition is significantly higher than that under the treatment of PU-WS13. We listed all statistical tests in the **Methods** (lines 790 to 800).

Reviewer #3 (Remarks to the Author):

The revised manuscript has appropriately addressed my central comments. I appreciate the clarification. I have no additional concerns.

Reviewer #4 (Remarks to the Author):

The authors have improved their manuscript substantially. It is strong work and merits publication in my view.

Reviewer #5 (Remarks to the Author):

Overall, this is an impressive scientific study. I will focus upon the machine learning and statistical analysis aspects of this manuscript as they comprise my domain of expertise. While the VSP analysis method itself is not unreasonable, the description of the method is misleading and inaccurate. The notations used are also somewhat confusing. I provide some recommendations below:

First, it is not that a "special type of GP" is used for geostatistics. The idea behind geostatistics is that the variable being modeled, which is denoted by the authors as Z_i , although I would recommend describing it as $Z(x_i, y_i)$ to denote the value of Z at x -coordinate (x_i) and y -coordinate (y_i) of the i -th location. What is going on in Ordinary Kriging is that the variable $Z(x, y)$ is modeled as: $Z(x, y) = \mu(x, y) + w(x, y) + e(x, y)$ for every point (x, y) , where $\mu(x, y) = \mu$ is a constant mean (customarily assumed in Ordinary Kriging), $w(x, y)$ is a latent spatial process that captures spatial associations and variations in the data, and $e(x, y)$ is a white noise process that captures micro-scale variation or measurement error variances.

It is the spatial process $w(x, y)$ that is crucial here. This process is modeled as a Gaussian process with a covariance function $C(h)$ that models the covariance between $Z(x_i+h, y_i+h)$ and $Z(x, y)$ where h is the displaced location. This is a typical stationary Gaussian process used in spatial statistics and the "gstat" R package the authors are using for their analysis essentially fits this spatial process model to their data.

My primary concern is with regard to statistically rigorous uncertainty quantification in the estimation of the model. The authors largely lie upon optimization and plug-in estimates to calculate inferential statistical measures such as spatial variance, Pearson's correlation and p -values to decide on significance of their findings. However, how efficiently the uncertainty quantification in their analysis propagates through the model and the noisy data is not clear from the analysis. The p -values calculated and presented are based on asymptotic assumptions for large sample sizes and the nature of spatial association that may not be correct.

I would encourage the authors to present a Bayesian data analysis either using the `spBayes` R package (<https://cran.r-project.org/package=spBayes>) or the `geoR` package (<https://cran.r-project.org/package=geoR>) to fit a Bayesian geostatistical model. In particular, the authors can use the `bayesGeostatExact()` function to fit an exact Bayesian geostatistical model (without resorting to more computationally expensive algorithms) by first fixing the ratio of the nugget to the spatial variance (using point estimates from a variogram analysis such as how the authors have already done). The authors can then interpolate the $Z(x, y)$ over the entire domain using the posterior predictive mean of Z at any point (x, y) . This will quantify uncertainty much better than is currently achieved and impact substantial scientific rigor to their data analysis.

References on Bayesian geostatistics:

Banerjee, S. , Carlin, B. P., and Gelfand, A. E. Hierarchical Modeling and Analysis for Spatial Data. Second Edition. CRC Press/Chapman & Hall.

See other references in the spBayes and geoR packages.

REVIEWER COMMENTS (Manuscript NCOMMS-23-14689B)

Reviewer #3 (Remarks to the Author):

The revised manuscript has appropriately addressed my central comments. I appreciate the clarification. I have no additional concerns.

Reviewer #4 (Remarks to the Author):

The authors have improved their manuscript substantially. It is strong work and merits publication in my view.

Reviewer #5 (Remarks to the Author):

Overall, this is an impressive scientific study. I will focus upon the machine learning and statistical analysis aspects of this manuscript as they comprise my domain of expertise. While the VSP analysis method itself is not unreasonable, the description of the method is misleading and inaccurate. The notations used are also somewhat confusing. I provide some recommendations below:

First, it is not that a "special type of GP" is used for geostatistics. The idea behind geostatistics is that the variable being modeled, which is denoted by the authors as Z_i , although I would recommend describing it as $Z(x_i, y_i)$ to denote the value of Z at x -coordinate (x_i) and y -coordinate (y_i) of the i -th location. What is going on in Ordinary Kriging is that the variable $Z(x, y)$ is modeled as: $Z(x, y) = \mu(x, y) + w(x, y) + e(x, y)$ for every point (x, y) , where $\mu(x, y) = \mu$ is a constant mean (customarily assumed in Ordinary Kriging), $w(x, y)$ is a latent spatial process that captures spatial associations and variations in the data, and $e(x, y)$ is a white noise process that captures micro-scale variation or measurement error variances. It is the spatial process $w(x, y)$ that is crucial here. This process is modeled as a Gaussian process with a covariance function $C(h)$ that models the covariance between $Z(x_i + h, y_i + h)$ and $Z(x, y)$ where h is the displaced location. This is a typical stationary Gaussian process used in spatial statistics and the "gstat" R package the authors are using for their analysis essentially fits this spatial process model to their data.

Response: Thanks for the suggestions. We have revised the description in the Methods from line 729 to 746 accordingly. Appropriate references have been added to reference list. All changes directly or indirectly related to reviewer concerns have been highlighted in yellow (original changes in response to first round of review) and green (responses to reviewer 5).

My primary concern is with regard to statistically rigorous uncertainty quantification in the estimation of the model. The authors largely lie upon optimization and plug-in estimates to calculate inferential statistical measures such as spatial variance, Pearson's correlation and p -values to decide on significance of their findings. However, how efficiently the uncertainty quantification in their analysis propagates through the model and the noisy data is not clear from the analysis. The p -values calculated and presented are based on asymptotic assumptions for large sample sizes and the nature of spatial association that may not be correct. I would encourage the authors to present a Bayesian data analysis either using the spBayes R package (<https://cran.r-project.org/package=spBayes>) or the geoR package (<https://cran.r-project.org/package=geoR>) to fit a Bayesian geostatistical model. In particular, the authors can use the bayesGeostatExact() function to fit an exact Bayesian geostatistical model (without resorting to more computationally expensive algorithms) by first fixing the ratio of the nugget to the spatial variance (using point estimates from a variogram analysis such as how the authors have already done). The authors can then interpolate the $Z(x, y)$ over

the entire domain using the posterior predictive mean of Z at any point (x,y) . This will quantify uncertainty much better than is currently achieved and impact substantial scientific rigor to their data analysis.

References on Bayesian geostatistics:

Banerjee, S. , Carlin, B. P., and Gelfand, A. E. Hierarchical Modeling and Analysis for Spatial Data. Second Edition. CRC Press/Chapman & Hall.

See other references in the spBayes and geoR packages.

Response: Thanks for the suggestions. We performed a comparison between Ordinary kriging and Bayesian kriging. The results are presented in the new Fig. S11. They are described in the Result section at lines 289-295 and lines 353-356, and also in the Methods section from line 808 to line 822. Appropriate references have been added to reference list. All changes directly or indirectly related to reviewer concerns have been highlighted in yellow (original changes in response to first round of review) and green (responses to reviewer 5).

Reviewer #5 (Remarks to the Author):

My outstanding concerns from the earlier round have been satisfactorily addressed. I have no further concerns.